# ADVERSARIAL PROBLEMS FOR GENERATIVE NETWORKS

## ABSTRACT

We are interested in the design of generative networks. The training of these mathematical structures is mostly performed with the help of adversarial (min-max) optimization problems. We propose a simple methodology for constructing such problems assuring, at the same time, consistency of the corresponding solution. We give characteristic examples developed by our method, some of which can be recognized from other applications and some are introduced here for the first time. We compare various possibilities by applying them to well known datasets using neural networks of different configurations and sizes.

## 1 INTRODUCTION

The problem we are interested in, can be summarized as follows: We are given two collections of training data $\{\mathbf{z}_j\}$ and $\{\mathbf{x}_i\}$. In the first set the samples follow the *origin* probability density $h(\mathbf{z})$ and in the second the *target* density $f(\mathbf{x})$. The target density $f(\mathbf{x})$ is considered unknown while $h(\mathbf{z})$ can either be known with the possibility to produce samples $\mathbf{z}_j$ every time it is necessary or unknown in which case we have a second fixed training set $\{\mathbf{z}_j\}$. Our goal is to design a deterministic transformation $G(\mathbf{z})$ so that the data $\{\mathbf{y}_j\}$ produced by applying the transformation $\mathbf{y} = G(Z)$ onto $\{\mathbf{z}_j\}$ follow the target density $f(\mathbf{y})$.

Of course one may wonder whether the proposed problem enjoys any solution, namely, whether there indeed exists a transformation $G(\mathbf{z})$ capable of transforming $\mathbf{z}$ into $\mathbf{y}$ with the former following the origin density $h(\mathbf{z})$ and the latter the target density $f(\mathbf{y})$. The problem of transforming random vectors has been analyzed in Box & Cox (1964) where *existence* is shown under general conditions. Computing, however, the actual transformation is a completely different challenge with one of the possible solutions relying on adversarial approaches applied to neural networks.

The most well known usage of this result is the possibility to generate synthetic data that follow the unknown target density $f(\mathbf{x})$. In this case $h(\mathbf{z})$ is selected to be simple (e.g. i.i.d. standard Gaussian or i.i.d. uniform) so that generating realizations from $h(\mathbf{z})$ is straightforward. As mentioned, the adversarial approach can be applied even if the origin density $h(\mathbf{z})$ is unknown provided that we have a dataset $\{\mathbf{z}_j\}$ with data following the origin density.

It was Goodfellow et al. (2014) that first introduced the idea of *adversarial* (min-max) optimization and demonstrated that it results in the determination of the desired transformation $G(\mathbf{z})$ (consistency). Alternative adversarial approaches were subsequently suggested by Martin Arjovsky & Bottou (2017); Bińkowski et al. (2018) and shown to also deliver the correct transformation $G(\mathbf{z})$.

We must mention the work of Nowozin et al. (2016) in which a class of min-max optimizations, f-GANs, was defined to design generator/discriminator pairs. Then, Liu et al. (2017) defined the adversarial divergences class of objective function which further combined f-GANs, MMD-GAN (Li et al., 2017), WGAN, WGAN-GP (Gulrajani et al., 2017), and entropic regularized optimal transport problems. Also, they investigated under what conditions the discriminator's class has the effect of matching generalized moments. Next, the work of Song & Ermon (2019) connected f-GANs and Wasserstein GANs (WGANs) (Martin Arjovsky & Bottou, 2017), and later Birrell et al. (2020) generalized the results by introducing the $(f, \Gamma)-$divergencies which allowed to bridge f-divergencies and integral probability metrics.

Our class of generative adversarial problems establishes a one-to-one correspondence with f-gans under the ideal (non data-driven) setup. However, we believe that our approach enjoys certain signif-

icant advantages: First, the definition of the two functions $\phi(z), \psi(z)$ in Equ. equation 8 is straightforward while Nowozin et al. (2016) requires to solve an additional optimization problem for the derivation of each GAN loss. An additional benefit of our approach is the complete control over the result of the maximization problem that defines the discriminator. In other words, we can decide what function, the discriminator must estimate. In Nowozin et al. (2016) such flexibility does not exist. This is important because we can select the approximation function properly so that we avoid the need to impose difficult constraints on the discriminator output (e.g. positivity) since such constraints tend to seriously affect the approximation quality of the corresponding neural network. Further, there is no need for the discriminator to be a Lipschitz function, as WGAN or WGAN-GP something that needs extra operations to ensure that the discriminator is Lipschitz.

Furthermore, we will show that the function the discriminator tries to approximate is a transformation of the likelihood ratio $r(\mathbf{x}) = g(\mathbf{x})/f(\mathbf{x})$ and there are important applications in Statistics where one is interested in estimating only the transformation of the likelihood ratio with the most common cases being the likelihood ratio itself, its logarithm (log-likelihood ratio), or the ratio $\frac{r(\mathbf{x})}{1+r(\mathbf{x})}$ which plays the role of the posterior probability between two densities. In other words, there are applications where one is interested only in the "max" part of the min-max problem.

Finally, because we know what transformation of the likelihood function the discriminator tries to approximate, it is possible to compare the different GANs on how closely they reach the optimal value of the likelihood ratio $r(\mathbf{x}) = 1$ meaning $f(\mathbf{x}) = g(\mathbf{x})$.

As in Nowozin et al. (2016), we will show that our methods provides an *abundance* of adversarial problems that are capable of identifying the appropriate transformation $G(\mathbf{z})$. Furthermore, we will also provide a simple recipe as to how we can successfully construct such problems.

Arguing along the same lines of the existing min-max formulations: We would like to optimally specify a vector transformation $G(\mathbf{z})$, the *generator*, and a scalar function $D(\mathbf{x})$, the *discriminator*. To achieve this, for each combination $\{G(\mathbf{z}), D(\mathbf{x})\}$ we define the cost function

$$J(G, D) = \mathbb{E}_{\mathbf{x} \sim f}\big[\phi\big(D(\mathbf{x})\big)\big] + \mathbb{E}_{\mathbf{z} \sim h}\big[\psi\big(D(G(\mathbf{z}))\big)\big] \tag{1}$$

where $\phi(z), \psi(z)$ are two scalar functions of the scalar $z$ and $\mathbb{E}_{\mathbf{x} \sim f}[\cdot], \mathbb{E}_{\mathbf{z} \sim h}[\cdot]$ denote expectation with respect to the density $f(\mathbf{x}), h(\mathbf{z})$ respectively. The optimum combination generator/discriminator is then identified by solving the following min-max problem

$$\min_{G(\mathbf{z})} \max_{D(\mathbf{x})} J(G, D) = \min_{G(\mathbf{z})} \max_{D(\mathbf{x})} \big\{ \mathbb{E}_{\mathbf{x} \sim f}\big[\phi\big(D(\mathbf{x})\big)\big] + \mathbb{E}_{\mathbf{z} \sim h}\big[\psi\big(D(G(\mathbf{z}))\big)\big] \big\}. \tag{2}$$

We must point out that our goal is not to solve equation 2, but rather *find a class of functions $\phi(z), \psi(z)$ so that the transformation $G(\mathbf{z})$ that will come out of the solution of equation 2 is such that $\mathbf{y} = G(\mathbf{z})$ follows the target density $f(\mathbf{y})$ when $\mathbf{z}$ follows the origin density $h(\mathbf{z})$.*

If $\mathbf{z}$ is random following $h(\mathbf{z})$ then $\mathbf{y} = G(\mathbf{z})$ is also random and we denote with $g(\mathbf{y})$ its corresponding probability density. Clearly, there exists a correspondence between transformations $G(\mathbf{z})$ and densities $g(\mathbf{y})$ when the density $h(\mathbf{z})$ of $\mathbf{z}$ is fixed. Since we can write

$$\mathbb{E}_{\mathbf{z} \sim h}\big[\psi\big(D(G(\mathbf{z}))\big)\big] = \mathbb{E}_{\mathbf{y} \sim g}\big[\psi\big(D(\mathbf{y})\big)\big],$$

this allows us to argue that the min-max problem in equation 2 is equivalent to

$$\min_{g(\mathbf{y})} \max_{D(\mathbf{x})} \big\{ \mathbb{E}_{\mathbf{x} \sim f}\big[\phi\big(D(\mathbf{x})\big)\big] + \mathbb{E}_{\mathbf{y} \sim g}\big[\psi\big(D(\mathbf{y})\big)\big] \big\} \tag{3}$$

It is now possible to combine the two expectations by applying a change of measure and a change of variables and equivalently write equation 3 as follows:

$$\min_{g(\mathbf{y})} \max_{D(\mathbf{x})} \big\{ \mathbb{E}_{\mathbf{x} \sim f}\big[\phi\big(D(\mathbf{x})\big)\big] + \int \psi\big(D(\boldsymbol{x})\big) \frac{g(\boldsymbol{x})}{f(\boldsymbol{x})} f(\boldsymbol{x}) d\boldsymbol{x} \big\} =$$

$$\min_{g(\mathbf{x})} \max_{D(\mathbf{x})} \big\{ \mathbb{E}_{\mathbf{x} \sim f}\big[\phi\big(D(\mathbf{x})\big)\big] + \mathbb{E}_{\mathbf{x} \sim f}\big[r(\mathbf{x})\psi\big(D(\mathbf{x})\big)\big] \big\} =$$

$$\min_{g(\mathbf{x})} \max_{D(\mathbf{x})} \mathbb{E}_{\mathbf{x} \sim f}\big[\phi\big(D(\mathbf{x})\big) + r(\mathbf{x})\psi\big(D(\mathbf{x})\big)\big] \big\}$$

where $r(\mathbf{x}) = g(\mathbf{x})/f(\mathbf{x})$ denotes the corresponding likelihood ratio. Since $f(\mathbf{x})$ is also fixed, there is again a correspondence between $r(\mathbf{x})$ and $g(\mathbf{x})$, hence the previous min-max problem becomes equivalent to

$$\min_{r(\mathbf{x}) \in \mathbb{L}_f} \max_{D(\mathbf{x})} \mathbb{E}_{\mathbf{x} \sim f}\big[\phi\big(D(\mathbf{x})\big) + r(\mathbf{x})\psi\big(D(\mathbf{x})\big)\big]. \tag{4}$$

Here $\mathbb{L}_f$ denotes the class of all likelihood ratios $r(\mathbf{x})$ with respect to the density $f(\mathbf{x})$, namely, all the functions $r(\mathbf{x})$ that satisfy

$$\mathbb{L}_f = \left\{ r(\mathbf{x}) : \ r(\mathbf{x}) \geq 0, \ \int r(\mathbf{x}) f(\mathbf{x}) \, d\mathbf{x} = 1 \right\}. \tag{5}$$

Using these definitions, let us define the cost

$$J(r, D) = \mathbb{E}_{\mathbf{x} \sim f} \left[ \phi(D(\mathbf{x})) + r(\mathbf{x}) \psi(D(\mathbf{x})) \right] \tag{6}$$

and, according to equation 4, we are interested in the following min-max problem

$$\min_{r(\mathbf{x}) \in \mathbb{L}_f} \max_{D(\mathbf{x})} J(r, D). \tag{7}$$

As mentioned, our actual goal is not to solve the adversarial problem. Instead, we would like to properly *identify* pairs of functions $\{\phi(z), \psi(z)\}$ so that equation 7 *accepts as solution the function* $r(X) = 1$. Indeed, if $r(X) = 1$ is the solution to equation 7, this means that $g(\mathbf{x}) = f(\mathbf{x})$ is the solution to equation 3 and, finally, that the optimum $G(\mathbf{x})$ obtained from equation 1 is such that $\mathbf{y} = G(\mathbf{x})$ follows $g(\mathbf{y}) = f(\mathbf{y})$ which, of course, is our original objective. Even though the min-max problem in equation 1 is what we attempt to solve, it is through equation 7 that we understand what its solution entails. In the next section we focus on equation 6, equation 7 and propose a simple design method (recipe) for the two functions $\phi(z), \psi(z)$ that assures that the solution of equation 7 is indeed $r(\mathbf{x}) = 1$. Before we discuss the details of our work, we would like to summarize this paper's contribution.

- We design a family of GANs problems using a likelihood ratio approach. In this class, all optimization problems have the desired property that the generator output follows the target distribution of the random vector of interest, $\mathbf{x}$, in other words, that the likelihood ratio of the two distributions is equal to one.

- We propose a straightforward *recipe* to explore the GANs family. With this methodology, we were able to identify subgroups characterized by specific transformations of the likelihood ratio. In these subgroups, we discovered novel objective functions and classified to them previously introduced GANs (such as Wasserstein, Cross-Entropy GANs).

- We propose a new *online* metric for evaluating the performance of our generative model during training.

- Our experiments provide insights for the behavior of the different GANs objective functions, with some of the novel objective functions performing better than the already known GANs.

## 2 A CLASS OF FUNCTIONS $\phi(z), \psi(z)$

Suppose that $\omega(r)$ is a *strictly increasing and (left and right) differentiable* scalar function of the nonnegative scalar $r$, i.e. $r \in [0, \infty)$. Denote with $\mathbb{J}_\omega = \omega([0, \infty))$ the range of values of $\omega(r)$ and let $\omega^{-1}(z)$ be the inverse function of $\omega(r)$ which exists and is defined for $z \in \mathbb{J}_\omega$. Let $\rho(z) > 0$ be a positive scalar function also defined for $z \in \mathbb{J}_\omega$ then, using $\omega(r)$ and $\rho(z)$, we propose the following pair $\phi(z), \psi(z)$

$$\phi'(z) = -\omega^{-1}(z)\rho(z), \ \ \psi'(z) = \rho(z), \tag{8}$$

where "$'$" denotes derivative. Since $\omega(r)$ and $\rho(z)$ are arbitrary (provided they satisfy the strict increase and positivity constraint respectively), the class of pairs defined by equation 8 is very rich allowing for a multitude of choices. We show next that *any* such pair $\{\phi(z), \psi(z)\}$ gives rise to a min-max problem, as in equation 7, that accepts $r(\mathbf{x}) = 1$ as its unique solution. We prove this claim in two steps. The first, involves a theorem where we consider a simplified version of the min-max problem.

**Theorem 1** *Let $\omega(r), \phi(z), \psi(z)$ and $\mathbb{J}_\omega$ be defined as above with the additional constraint $\psi(\omega(1)) = 0$. Fix $r \geq 0$ and consider $\phi(D) + r\psi(D)$ as a function of the scalar $D$. Then, for any $D \in \mathbb{J}_\omega$, we have that*

$$\phi(D) + r\psi(D) \leq \phi(\omega(r)) + r\psi(\omega(r)), \tag{9}$$

*with equality if and only if $D = \omega(r)$.*

*Consider next the minimization with respect to $r$ of the maximal value in equation 9. It is then true that*

$$\min_{r \geq 0} \left\{ \phi\big(\omega(r)\big) + r\psi\big(\omega(r)\big) \right\} = \phi\big(\omega(1)\big), \tag{10}$$

*with equality if and only if $r = 1$.*

A consequence of Theorem 1 is the next corollary, which constitutes the second and final step in proving that the adversarial problem defined in equation 7 has as unique solution the function $r(X) = 1$.

**Corollary 1** *If the functions $\phi(z), \psi(z)$ satisfy equation 8 and $\omega(r)$ is strictly increasing and left and right differentiable, then in the adversarial problem defined in equation 7 the maximizer is $D(X) = \omega\big(r(X)\big)$ and the minimizer is $r(X) = 1$, while the resulting min-max value is equal to*

$$\min_{r(X) \in \mathbb{L}_f} \max_{D(\mathbf{x})} \mathbb{E}_{\mathbf{x} \sim f} \left[ \phi\big(D(\mathbf{x})\big) + r(X)\psi\big(D(\mathbf{x})\big) \right] = \phi\big(\omega(1)\big) + \psi\big(\omega(1)\big). \tag{11}$$

## 2.1 SUBCLASSSES OF THE GANS FAMILY

Let us now present some of the subclasses of the GANs family, where each subclass is characterized by the type of transformation of the likelihood ratio, $\omega(r)$. Once we fix $\omega(r)$ and give pairs $\{\phi(z), \psi(z)\}$ that satisfy equation 8, we are able to "pick" objective functions laying in the $\omega(r)$ subregion.

**Subclass A**: $\omega(r) = r^\alpha$   The first examined subclass is the simplest one, consisting of just powers of the likelihood ratio. To the best of our knowledge, this is the first work proposing objective functions from this class. To find the pairs $\{\phi(z), \psi(z)\}$ we proceed as follows.

We have that $\omega^{-1}(z) = z^{\frac{1}{\alpha}}$ and $\mathbb{J}_\omega = [0, \infty)$. According to equation 8, for $z \in [0, \infty)$ we must define $\phi'(z) = -z^{\frac{1}{\alpha}}\rho(z)$, $\psi'(z) = \rho(z)$. Also $r = D^{-a}$. Some examples of this subclass are presented in Table 1.

For the particular selection $\omega(r) = r$ (corresponding to $\alpha = 1$) we can show that the resulting cost is equivalent to the Bregman cost (Bregman, 1967). In fact there is a one-to-one correspondence between our $\rho(z)$ function and the function that defines the Bregman cost. This correspondence however is lost once we switch to a different $\alpha$ or a different $\omega(r)$ function, suggesting that the proposed class of pairs $\{\phi(z), \psi(z)\}$, is far richer than the class induced by the Bregman cost.

Table 1: Subclass A optimization problems for GANs

| **GAN** | $\phi(z)$ | $\psi(z)$ | $\mathbb{J}_\omega$ |
|---|---|---|---|
| A1a | $-z$ | $\log(z)$ | $[0, \infty)$ |
| A1b | $-\log(z)$ | $-z^{-1}$ | $[0, \infty)$ |
| A2 | $-(1 + z)$ | $-(1 + z^{-1})$ | $[0, \infty)$ |
| A3 | $-\log(1 + z)$ | $-\log(1 + z^{-1})$ | $[0, \infty)$ |
| MSE | $-0.5z^2$ | $z$ | $[0, \infty)$ |

**Subclass B**: $\omega(r) = \alpha^{-1} \log r$ This subclass considers one of the most popular transformations of the likelihood ratio, the log-likelihood ratio. As with subclass A for the first time, the next examples are presented. They can be used either under a min-max setting, for the determination of the generator/discriminator pair, or under a pure maximization setting for the direct estimation of the log-likelihood ratio function $\log r(\mathbf{x})$.

We have $\omega^{-1}(z) = e^{\alpha z}$ and $\mathbb{J}_\omega = \mathbb{R}$. As before $\rho(z)$ must be strictly positive and, according to equation 8, for all real $z$ we must define $\phi'(z) = -e^{\alpha z}\rho(z)$, $\psi'(z) = \rho(z)$. And the likelihood ratio is $r = e^{aD}$. In Table 2 we can see some examples of this subclass.

**Subclass C**: $\omega(r) = \frac{r}{r+1}$ As we already mentioned, this is another important transform of the likelihood ratio. Interestingly, in this subclass belongs the first introduced GAN (Goodfellow et al., 2014) the *Cross Entropy* GAN.

Table 2: Subclass B optimization problems for GANs

| **GAN** | $\phi(z)$ | $\psi(z)$ | $\mathbb{J}_\omega$ |
|---|---|---|---|
| B1a | $-e^z$ | $e^z$ | $\mathbb{R}$ |
| B1b | $-z$ | $-e^{-z}$ | $\mathbb{R}$ |
| Exponential | $-e^{0.5z}$ | $-e^{-0.5z}$ | $\mathbb{R}$ |
| B2 | $-\log(1+e^z)$ | $-\log(1+e^{-z})$ | $\mathbb{R}$ |

When $\omega(r) = \frac{r}{r+1}$ we have $\omega^{-1}(z) = \frac{z}{1-z}$ and $\mathbb{J}_\omega = [0,1]$. For $\rho(z) > 0, z \in [0,1]$ we must define the functions $\phi(z), \psi(z)$ according to equation 8 $\phi'(z) = -\frac{z}{1-z}\rho(z)$, $\psi'(z) = \rho(z)$. In this case the likelihood ratio is $r = \frac{D}{1-D}$. In Table 3 we see the cross entropy GAN and C2 which is presented here for the first time.

Table 3: Subclass C optimization problems for GANs

| **GAN** | $\phi(z)$ | $\psi(z)$ | $\mathbb{J}_\omega$ |
|---|---|---|---|
| Cross Entropy | $\log(1-z)$ | $\log(z)$ | $[0,1]$ |
| C2 | $z + \log(1-z)$ | $z$ | $[0,1]$ |

**Subclass D**: $\omega(r) = \text{sign}(\log r)$ This is a special case of $\omega(r)$ with the corresponding function not being strictly increasing. It turns out that we can still come up with optimization problems, two of which are known and used in practice, by considering $\omega(r)$ as a *limit* of a sequence of strictly increasing functions.

*Monotone Loss:* As a first approximation we propose $\text{sign}(z) \approx \tanh(\frac{c}{2}z)$ where $c > 0$ a parameter. We note that $\lim_{c\to\infty} \tanh(\frac{c}{2}z) = \text{sign}(z)$. Using this approximation we can write

$$\text{sign}(\log r) \approx \tanh\left(\frac{c}{2}\log r\right) = \frac{r^c - 1}{r^c + 1} = \omega(r). \tag{12}$$

As we mentioned, we have exact equality for $c \to \infty$. Let us perform our analysis by assuming that $c$ is finite. We note that $\omega^{-1}(z) = (\frac{1+z}{1-z})^{\frac{1}{c}}$ and $\mathbb{J}_\omega = [-1, 1]$. Consequently, if $\rho(z) > 0$ for $z \in [-1, 1]$, we must define $\phi'(z) = -\left(\frac{1+z}{1-z}\right)^{\frac{1}{c}}\rho(z)$, $\psi'(z) = \rho(z)$.

D1) If we let $c \to \infty$ in order to converge to the desired sign function, this yields $\phi'(z) = -\rho(z)$ and $\psi'(z) = \rho(z)$. This suggests that $\phi(z) = -\int^z \rho(x)dx$ is decreasing and $\psi(z) = \int^z \rho(x)dx = -\phi(z)$ is increasing. In fact any strictly increasing function $\psi(z)$ can be adopted provided we select $\phi(z) = -\psi(z)$.

There is a popular combination that falls under Case D1). In particular, the selection $\psi(z) = z = -\phi(z)$ reminds us of Wasserstein GAN Martin Arjovsky & Bottou (2017), with two differences, in our case $z$ should lie in $[-1, 1]$ and the discriminator is not constrained to be a Lipschitz function.

*Hinge Loss:* As a second approximation we use the expression $\text{sign}(z) \approx \text{sign}(z)|z|^{\frac{1}{c}}$, $c > 0$, which is strictly increasing, continuous and converges to $\text{sgn}(z)$ as $c \to \infty$. This suggests that

$$\text{sign}(\log r) \approx \text{sign}(\log r)|\log r|^{\frac{1}{c}} = \omega(r), \tag{13}$$

and $\omega^{-1}(z) = e^{z^c}$. Since $\omega(r)$ can assume any real value we conclude that $\mathbb{J}_\omega = \mathbb{R}$ which, clearly, differs from the previous approximation where we had $\mathbb{J}_\omega = [-1, 1]$. If $\rho(z) > 0, z \in \mathbb{R}$ then, according to equation 8 we must define $\phi'(z) = -e^{z^c}\rho(z)$, $\psi'(z) = \rho(z)$. We present the following case that leads to a very well known pair from a completely different application.

D2) If we select $\psi'(z) = \rho(z) = \{e^{-|z|^{\frac{1}{c}}} + \mathbf{1}_{z<-1}\} > 0$ then $\phi'(z) = -e^{z^{\frac{1}{c}}}\{e^{-|z|^{\frac{1}{c}}} + \mathbf{1}_{z<-1}\}$. If we now let $c \to \infty$, we obtain the limiting form for the derivatives which become $\psi'(z) = -\mathbf{1}_{z<1}$ and $\phi'(z) = \mathbf{1}_{z>-1}$. By integrating we arrive at $\phi(z) = -\max\{1+z, 0\}$ and $\psi(z) = -\max\{1-z, 0\}$. we notice that since $c \to \infty$ we cannot find the likelihood ratio in terms of the discriminator. The cost based on this particular pair is called the *hinge loss* Tang (2013) and it is very popular in binary classification where one is interested only in the maximization problem. The corresponding method is known to exhibit an overall performance which in practice is considered among the best Rosasco et al. (2004); Janocha & Czarnecki (2017). Here, as in Zhao et al. (2016), we propose the hinge loss as a means to perform adversarial optimization for the design of the generator $G(\mathbf{x})$.

Table 4: Optimization problems for GANs

| **GAN** | $\phi(z)$ | $\psi(z)$ | $\mathbb{J}_\omega$ |
|---|---|---|---|
| Hinge | $-(1+z)_+$ | $-(1-z)_+$ | $\mathbb{R}$ |
| Wasserstein | $z$ | $-z$ | $\mathbb{R}$ |

In Table 4 we can see Hinge and Wasserstein GANs optimization problems.

The detailed derivation of the above optimization problems and some more examples can be found in Appendix A.2.

This completes our presentation of examples. However, we must emphasize, that these are only a few illustrations of possible pairs $\{\phi(z), \psi(z)\}$ one can construct. Indeed combining, as dictated by equation 8, any strictly increasing function $\omega(r)$ with any positive function $\rho(z)$ generates a legitimate pair $\{\phi(z), \psi(z)\}$ and a corresponding min-max problem equation 7 that enjoys the desired solution $r(\mathbf{x}) = 1$.

**Remark 1** *Given the numerous choices we have in defining adversarial optimizations for solving the same problem, one may wonder whether there exists a means to rank these methods and identify the most efficient, at least for classes of data. This is an extremely challenging question which, unfortunately, finds no answer even in simpler problems. For example, in binary classification there are also classes of optimization problems that are used to train neural networks Bartlett et al. (2006); Masnadi-Shirazi & Vasconcelos (2009). However, until now, no theoretical analysis exists that can order them and designate the most efficient one. This is possible only through experience with countless simulations.*

## 3 DATA-DRIVEN SETUP AND NEURAL NETWORKS

Let us now consider the data-driven version of the problem. As mentioned, the target density $f(\mathbf{x})$ is unknown. Instead we are given a collection of realizations $\{\mathbf{x}_i\}$ that follow $f(\mathbf{x})$ and a second collection $\{\mathbf{z}_j\}$ that follows the origin density $h(\mathbf{z})$. These data constitute our *training set*. Regarding the second set $\{\mathbf{z}_j\}$ it can either become available "on the fly" when $h(\mathbf{z})$ is known by generating realizations every time they are needed, or it can be considered fixed from the start exactly as $\{\mathbf{x}_i\}$, if $h(\mathbf{z})$ is also unknown. As we pointed out in Section 1, we are interested in designing a generator $G(\mathbf{z})$ so that when we apply it onto the data $\mathbf{z}_j$, that is, $\mathbf{y}_j = G(\mathbf{z}_j)$ the resulting $\mathbf{y}_j$ will follow a density that matches the target density $f(\mathbf{x})$. Since we are now considering the data-driven version of the problem, we are going to limit $G(\mathbf{z}), D(\mathbf{x})$ to be the outputs of corresponding neural networks. Therefore the generator is replaced by $G(\mathbf{z}, \theta)$ while the discriminator by $D(\mathbf{x}, \vartheta)$ where $\theta, \vartheta$ summarize the parameters of the two neural networks. Of course instead of neural networks one could use any other parametric family, as SVMs, capable of efficiently approximating any nonlinear function.

Once we have selected our favorite $\omega(r)$ and $\rho(z)$ functions we can compute from equation 8 the functions $\phi(z), \psi(z)$ that enter into the min-max problem defined in equation 2. This problem, after limiting the generator and discriminator to neural networks, can be rewritten as follows

$$\min_\theta \max_\vartheta \mathcal{J}(\theta, \vartheta) = \min_\theta \max_\vartheta \left\{ \mathbb{E}_{\mathbf{x} \sim f} \left[ \phi\big(D(\mathbf{x}, \vartheta)\big) \right] + \mathbb{E}_{\mathbf{z} \sim h} \left[ \psi\big(D(G(\mathbf{z}, \theta), \vartheta)\big) \right] \right\}. \quad (14)$$

If $\theta_o, \vartheta_o$ are the corresponding optimum parameter values, and the structure of the two networks is sufficiently rich, we expect that $G(\mathbf{z}, \theta_o), D(\mathbf{x}, \vartheta_o)$ will approximate the optimum functions $D(\mathbf{x}), G(\mathbf{z})$ of the ideal problem in equation 2 respectively. In particular for $\theta_o$, the generator $G(\mathbf{z}, \theta_o)$, whenever applied onto any $\mathbf{z}_j$ that follows $h(\mathbf{z})$, it will result in a $\mathbf{y}_j = G(\mathbf{z}_j, \theta_o)$ that follows a density which is expected to be close to the target density $f(\mathbf{y})$.

**Remark 2** *When replacing $D(\mathbf{x}), G(\mathbf{z})$ with neural networks we must take special care of the corresponding outputs. Basically, we must guarantee that they are of the correct form. This is particularly important in the case of the scalar output $D(\mathbf{x}, \vartheta)$ of the discriminator. We recall that the optimum discriminator is $D(\mathbf{x}) = \omega\big(r(\mathbf{x})\big)$. This implies that we need to assure that $D(\mathbf{x}, \vartheta)$ takes values in $\mathbb{J}_\omega$ (the range of $\omega(r)$). Consequently, we must apply the proper nonlinearity in the output of the discriminator that will guarantee this fact.*

## 4 EXPERIMENTS

In this section, we want to examine the performance of the GANs objectives presented in Table 4 for different datasets. For that reason we tested their performance on four different datasets, namely MNIST (LeCun et al., 1998), CelebA (Liu et al., 2015), CIFAR-10 datasets (Krizhevsky et al., 2009), and Stanford Cars (Krause et al., 2013). We recall that GANs are notorious for their nonrobust behavior Bengio (2012); Creswell et al. (2018); Mescheder et al. (2017). For the stabilization of the training process, we used the maximum gradient-penalty methodology which was generalized to a class of Lipschitz GANs in Zhou et al. (2019) (implementation details in Appendix A.3). In this section we present our results for the regularization parameter $\lambda = 10$, in the Appendix we included results for different values of $\lambda$.

In Tables 5, 6 we present the final attained Frechet Inception Distances (FID) Heusel et al. (2017) and Kernel Inception Distances (KID) Bińkowski et al. (2018) scores after training. In Table 7 we computed the absolute difference of the discriminator estimated likelihood ratio from the optimal likelihood ratio, which is equal to one and the variance around the optimal value. For Hinge and Wasserstein GANs we cannot compute the likelihood ratio related metrics, as we mentioned in section 2.1. Also, in Figure 4 we show the evolution of FID (second row) and KID (third row) during training. In the first row we see the value of the distance $d(\mathbf{x}, \mathbf{y}) = |\mathbb{E}_{\mathbf{x} \sim f}[D(\mathbf{x})] - \mathbb{E}_{\mathbf{y} \sim g}[D(\mathbf{y})]|$. For the expectations estimation, we used 64 examples for MNIST and 128 for Stanford Cars, CelebA, and CIFAR-10. We believe that this is an insightful quantity since its value is an indicator of how accurate the discriminator can distinguish samples from the target distribution and the generator output. Finally, in Appendix A.3, some generated samples of our trained generative models are included.

Table 5: FID Scores

| GAN | CARS | CELEBA | CIFAR10 | MNIST |
|---|---|---|---|---|
| A1a | $23.30 \pm 0.10$ | $7.53 \pm 0.02$ | $9.67 \pm 0.01$ | $2.18 \pm 0.01$ |
| A1b | $24.22 \pm 0.22$ | $7.68 \pm 0.04$ | $9.69 \pm 0.04$ | $2.13 \pm 0.01$ |
| A2 | $24.40 \pm 0.04$ | $7.62 \pm 0.04$ | $9.53 \pm 0.04$ | $2.17 \pm 0.01$ |
| A3 | $23.64 \pm 0.23$ | $8.50 \pm 0.06$ | $9.72 \pm 0.04$ | $2.15 \pm 0.01$ |
| MSE | $23.99 \pm 0.04$ | $7.66 \pm 0.07$ | $9.61 \pm 0.03$ | $2.15 \pm 0.01$ |
| B1a | $23.60 \pm 0.12$ | $8.06 \pm 0.02$ | $9.67 \pm 0.05$ | $2.13 \pm 0.01$ |
| B1b | $23.91 \pm 0.07$ | $8.07 \pm 0.06$ | $9.67 \pm 0.04$ | $2.18 \pm 0.01$ |
| Exponential | $23.52 \pm 0.06$ | $9.15 \pm 0.03$ | $9.79 \pm 0.03$ | $2.13 \pm 0.01$ |
| B2 | $23.46 \pm 0.12$ | $9.39 \pm 0.06$ | $9.79 \pm 0.07$ | $2.09 \pm 0.01$ |
| Cross Entropy | $25.39 \pm 0.16$ | $7.53 + 0.03$ | $9.72 \pm 0.09$ | $2.08 \pm 0.01$ |
| C2 | $24.36 \pm 0.12$ | $7.47 \pm 0.01$ | $9.75 \pm 0.03$ | $2.10 \pm 0.01$ |
| Hinge | $22.88 \pm 0.14$ | $10.91 \pm 0.05$ | $9.99 \pm 0.03$ | $2.16 \pm 0.01$ |
| Wasserstein | $23.99 \pm 0.04$ | $10.08 \pm 0.04$ | $9.74 \pm 0.02.$ | $2.16 \pm 0.01$ |

Table 6: KID Scores

| GAN | CARS $\times 10^{-6} \pm \times 10^{-12}$ | CELEBA $\times 10^{-6} \pm \times 10^{-12}$ | CIFAR10 $\times 10^{-6} \pm \times 10^{-12}$ | MNIST $\times 10^{-4} \pm \times 10^{-8}$ |
|---|---|---|---|---|
| A1a | $4.03 \pm 2.45$ | $2.57 \pm 5.05$ | $2.75 \pm 8.47$ | $7.80 \pm 5.86$ |
| A1b | $3.33 \pm 1.89$ | $2.67 \pm 8.35$ | $2.36 \pm 5.86$ | $6.30 \pm 4.32$ |
| A2 | $4.36 \pm 3.18$ | $3.38 \pm 9.68$ | $3.09 \pm 8.35$ | $8.79 \pm 4.65$ |
| A3 | $4.42 \pm 2.49$ | $7.29 \pm 5.48$ | $2.40 \pm 4.95$ | $8.53 \pm 7.35$ |
| Mean Square | $3.79 \pm 1.71$ | $2.55 \pm 9.34$ | $2.09 \pm 5.63$ | $8.64 \pm 7.51$ |
| B1a | $6.17 \pm 3.62$ | $5.56 \pm 9.78$ | $2.35 \pm 5.52$ | $8.05 \pm 3.30$ |
| B1b | $5.24 \pm 2.27$ | $7.32 \pm 1.18$ | $2.63 \pm 8.42$ | $6.85 \pm 3.71$ |
| Exponential | $5.49 \pm 2.86$ | $10.05 \pm 6.10$ | $2.81 \pm 8.34$ | $6.88 \pm 3.41$ |
| B2 | $7.06 \pm 3.40$ | $12.48 \pm 7.38$ | $2.47 \pm 5.42$ | $8.82 \pm 3.69$ |
| Cross Entropy | $13.09 \pm 8.95$ | $3.53 \pm 12.65$ | $2.53 \pm 5.58$ | $6.16 \pm 5.37$ |
| C2 | $8.45 \pm 2.92$ | $3.11 \pm 9.72$ | $4.17 \pm 11.03$ | $7.72 \pm 3.36$ |
| Hinge | $4.97 \pm 3.50$ | $26.65 \pm 16.81$ | $3.97 \pm 9.42$ | $8.33 \pm 4.16$ |
| Wasserstein | $4.92 \pm 2.70$ | $20.45 \pm 28.38$ | $1.74 \pm 5.22$ | $7.33 \pm 3.33$ |

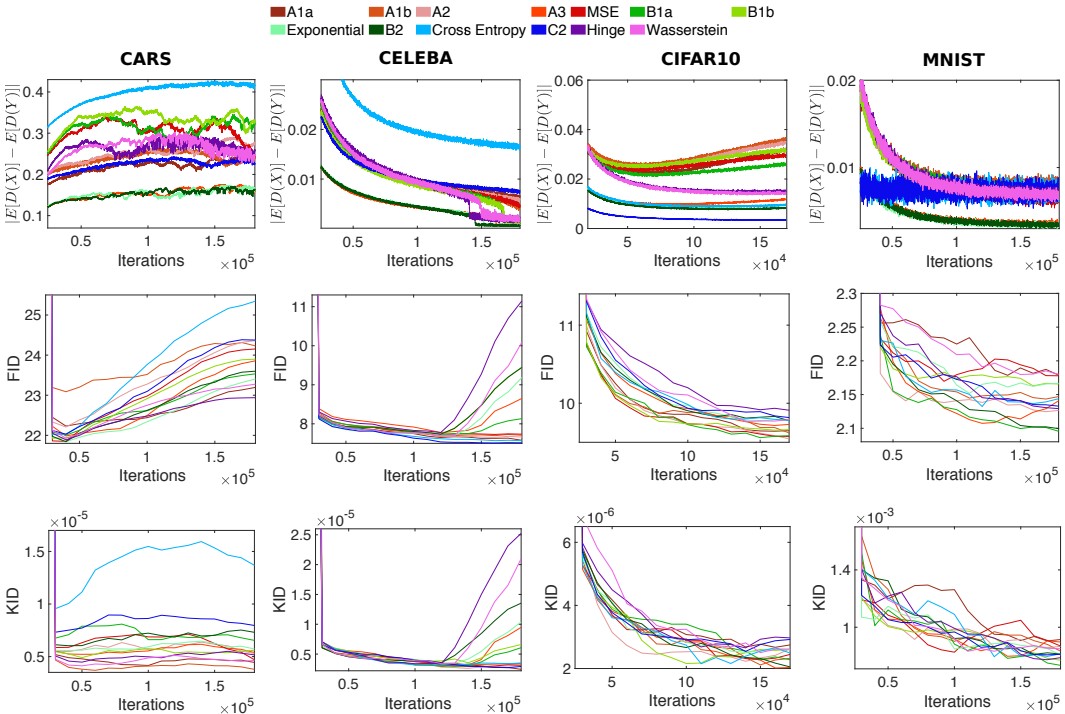

Figure 1: Evolution of the corresponding FID and KID scores during training for the datasets Stanford Cars, CelebA, CIFAR10, and MNIST.

Table 7: Mean absolute difference of the discriminator estimated likelihood ratio from the optimal value and variance around the optimal likelihood ratio.

| GAN | CARS $\times 1, \times 10^{-2}$ | CELEBA $\times 1, \times 10^{-3}$ | CIFAR10 $\times 10^{-2}, \times 10^{-4}$ | MNIST $\times 10^{-1}, \times 10^{-5}$ |
|---|---|---|---|---|
| A1a | $0.10, 0.29$ | $0.01, 0.17$ | $1.79, 1.33$ | $0.07, 6.49$ |
| A1b | $0.38, 0.37$ | $0.01, 0.20$ | $5.71, 1.73$ | $0.13, 8.05$ |
| A2 | $0.12, 0.21$ | $0.01, 0.11$ | $1.83, 1.16$ | $0.07, 5.25$ |
| A3 | $0.17, 0.13$ | $0.01, 0.14$ | $1.27, 4.79$ | $0.05, 2.72$ |
| Mean Square | $0.13, 0.62$ | $0.01, 0.26$ | $1.78, 2.04$ | $0.07, 6.16$ |
| B1a | $0.17, 0.72$ | $0.01, 0.49$ | $1.46, 1.36$ | $0.08, 8.91$ |
| B1b | $0.14, 0.40$ | $0.01, 0.57$ | $1.63, 1.49$ | $0.07, 7.67$ |
| Exponential | $0.03, 0.16$ | $0.01, 0.97$ | $0.52, 4.34$ | $0.04, 2.88$ |
| B2 | $0.03, 0.14$ | $0.01, 0.90$ | $0.55, 4.86$ | $0.04, 3.07$ |
| Cross Entropy | $0.16, 3.98$ | $0.03, 0.02$ | $0.52, 0.44$ | $1.39, 2.66$ |
| C2 | $0.41, 6.70$ | $0.03, 0.94$ | $0.38, 0.20$ | $1.39, 2.74$ |

For the MNIST dataset, the different GANs have very similar performance, something that is reasonable since this is a simple dataset. The $d(\mathbf{x}, \mathbf{y})$ curves are quickly converging indicating that the discriminator has stabilized. Interestingly, the FID, KID curves for GANs with similar $d(\mathbf{x}, \mathbf{y})$ have also close FID, KID scores. In particular, Subclass C GANs (Cross-Entropy with C2); Subclass D (Hinge) with Wasserstein; B2, B1a and A3; B1b with Exponential; A1b with A2; and A1a with MSE. For the CIFAR-10 dataset, the FID and KID scores for different objective functions tend to have similar behavior, but some differences exist. In particular, the objectives A1a, A1b, A2, MSE, B1a, B1b in the initial iterations create a steep valley for the metric $d(\mathbf{x}, \mathbf{y})$; and at the same time in the FID score, these objectives attain smaller FID scores faster than the other GANs. Also, the further improvement of the scores during training seems to be related to the behavior of $d(\mathbf{x}, \mathbf{y})$. For instance, the $d(\mathbf{x}, \mathbf{y})$ of A1b, A2 is still increasing during the last iterations, and the generated image quality (FID score) is improving. Lastly, similar to MNIST, GANs that have very close $d(\mathbf{x}, \mathbf{y})$ tend to have close FID, KID curves.

For Stanford Cars and CelebA, we observe some performance gaps between different subclasses. Specifically for CelebA, the Subclass D and Subclass B objective functions start to diverge after some training steps, something that is evident from the FID and KID curves where the scores increase dramatically after approximately the first half of iterations. Also, this is noticeable for the $d(x, y)$, where a sudden drop is evident around the same period. Interestingly the variance around the optimal likelihood ratio value is increased for the GANS who started to diverge. For Stanford cars, the Subclass C has the worst performance, with the recorded MMD score being an order larger than the other objectives (expect Subclass B –B2, Exponential objectives–). Furthermore, for this dataset, the distances $d(\mathbf{x}, \mathbf{y})$ are "noisy" (strong fluctuations) and with approximately ten times larger magnitude value when compared to the corresponding curves in the other datasets. It is reasonable since this dataset is a complex one, where we have natural scenes with cars, and, importantly, significantly smaller ($\sim 8000$ training samples, when MNIST, CIFAR10 has 60000 and CelebA 200000).

In summary, our simulations indicate that the Subclass A objectives A1a, A1b, A2, MSE have the best performance both in terms of the computed metrics (hence image generation quality) and stability during training. They might not give the exact best score in all datasets, but are very close to the best recorded, and, most importantly, they have an overall more stable behavior when compared to objectives from the other subclasses. Furthermore, the discriminator convergence to the optimal likelihood ratio between the dataset and the generator output can be an ad-hoc measure of the behavior/stability of our generative models than can be used during training to provide online, useful insights of the current training condition.

## 5 CONCLUSION

In this paper, we provided and demonstrated a straightforward methodology to determine loss functions that solve the generative adversarial problem. Our results suggest that there is not a single loss function that achieves the best performance in terms of the examined metrics for all different datasets. This performance variation among loss functions becomes evident as the increasing complexity of the datasets that convolutes the generation task is better addressed by some loss functions that clearly outperform others. Specifically, in simpler datasets, such as MNIST, the evaluated loss functions yield very similar performance, whereas, in more intricate datasets like CelebA, CIFAR-10, and Stanford Cars, performance "gaps" between the different loss functions, and different subclasses, emerges. Our findings also propose that in every generation task unexplored loss functions outperformed the previously proposed ones. Consequently, this function class is worth-exploring to identify new loss functions that can be used and evaluated in different applications. Our method provides a versatile tool that can be exploited in that direction.

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

# A APPENDIX

## A.1 PROOFS

**Theorem 1 Proof.** We note that the constraint $\psi\big(\omega(1)\big) = 0$ does not affect the generality of our class of functions since from equation 8 we have that $\psi(z)$, after integration, is defined up to an arbitrary additive constant. We can always select this constant so that the constraint is satisfied. We would also like to emphasize that this constraint is needed only for the proof of this theorem and it is not necessary for the corresponding min-max problem defined in equation 7.

For fixed $r$, to find the maximum of $\phi(D) + r\psi(D)$ we consider the derivative with respect to $D$ which, using equation 8, takes the form

$$\phi'(D) + r\psi'(D) = \big(r - \omega^{-1}(D)\big)\rho(D).$$

The strict increase of $\omega(r)$ is inherited by its inverse function $\omega^{-1}(z)$ which, combined with the positivity of $\rho(z)$, implies that the previous expression has the same sign as $r - \omega^{-1}(D)$ or $\omega(r) - D$. Consequently $D = \omega(r)$ is the *only critical point* of $\phi(D) + r\psi(D)$ which is a global maximum. Of course there are possibilities for extrema at the two end points of $\mathbb{J}_\omega$ but they can only be (local) minima.

Let us now focus on the resulting function $\phi\big(\omega(r)\big) + r\psi\big(\omega(r)\big)$. Taking its derivative with respect to $r$ yields

$$\big\{\phi\big(\omega(r)\big) + r\psi\big(\omega(r)\big)\big\}' = \big\{\phi'\big(\omega(r)\big) + r\psi'\big(\omega(r)\big)\big\}\,\omega'(r) + \psi\big(\omega(r)\big) = \psi\big(\omega(r)\big),$$

where the last equality is due to the specific definition of the two functions $\phi(z), \psi(z)$ in equation 8. Since $\psi'(z) = \rho(z) > 0$, this implies that $\psi(z)$ is strictly increasing, being also the integral of $\rho(z)$ it is continuous in $z$. If we combine this property with the strict increase and continuity (as a result of left and right differentiability) of $\omega(r)$ we conclude that $\psi\big(\omega(r)\big)$ is also strictly increasing and continuous in $r$. We recall that $\psi(z)$ is selected to satisfy $\psi\big(\omega(1)\big) = 0$, consequently for $r = 1$ the function $\phi\big(\omega(r)\big) + r\psi\big(\omega(r)\big)$ has a unique minimum which is global and no other critical points. Of course it can still exhibit extrema at $r = 0$ and/or $r \to \infty$ but they can only be (local) maxima. ∎

**Corollary 1 Proof.** First, we observe that

$$\mathbb{E}_{\mathbf{x}\sim f}\big[\phi\big(D(\mathbf{x})\big) + r(\mathbf{x})\psi\big(D(\mathbf{x})\big)\big] = \mathbb{E}_{\mathbf{x}\sim f}\big[\phi\big(D(\mathbf{x})\big) + r(\mathbf{x})\tilde{\psi}\big(D(\mathbf{x})\big)\big] + \psi\big(\omega(1)\big) \qquad (15)$$

with the last equality being true since $\mathbb{E}_{\mathbf{x}\sim f}[r(\mathbf{x})] = 1$ and where $\tilde{\psi}(z) = \psi(z) - \psi\big(\omega(1)\big)$. We start with the maximization problem. Since $D(\mathbf{x})$ is a function of $\mathbf{x}$ we have

$$\max_{D(\mathbf{x})} \mathbb{E}_{\mathbf{x}\sim f}\big[\phi\big(D(\mathbf{x})\big) + r(\mathbf{x})\tilde{\psi}\big(D(\mathbf{x})\big)\big] = \mathbb{E}_{\mathbf{x}\sim f}\Big[\max_{D(\mathbf{x})}\big\{\phi\big(D(\mathbf{x})\big) + r(\mathbf{x})\tilde{\psi}\big(D(\mathbf{x})\big)\big\}\Big]. \qquad (16)$$

The maximization under the expectation can be performed for each fixed $\mathbf{x}$. However, when we fix $\mathbf{x}$ then $r(\mathbf{x})$ becomes a constant and the result of the maximization depends only on the actual value of $r(\mathbf{x})$. This suggests that we can limit ourselves to functions of the form $D(\mathbf{x}) = D\big(r(\mathbf{x})\big)$. After this observation we can drop the dependence on $\mathbf{x}$ and perform, equivalently, the maximization $\max_D \big\{\phi\big(D(r)\big) + r\tilde{\psi}\big(D(r)\big)\big\}$ for each fixed $r$. The pair $\{\phi(z), \tilde{\psi}(z)\}$ satisfies the assumptions of Theorem 1, therefore maximization is achieved for $D(r) = \omega(r)$. This implies that

$$\max_{D(\mathbf{x})} \mathbb{E}_{\mathbf{x}\sim f}\big[\phi\big(D(\mathbf{x})\big) + r(\mathbf{x})\psi\big(D(\mathbf{x})\big)\big] = \mathbb{E}_{\mathbf{x}\sim f}\big[\phi\big(\omega\big(r(\mathbf{x})\big)\big) + r(\mathbf{x})\tilde{\psi}\big(\omega\big(r(\mathbf{x})\big)\big)\big] + \psi\big(\omega(1)\big).$$

We can now continue in a similar way for the minimization problem. Specifically

$$\min_{r(\mathbf{x})\in\mathbb{L}_f} \max_{D(\mathbf{x})} \mathbb{E}_{\mathbf{x}\sim f}\left[\phi\big(D(\mathbf{x})\big) + r(\mathbf{x})\tilde{\psi}\big(D(\mathbf{x})\big)\right] =$$

$$\min_{r(\mathbf{x})\in\mathbb{L}_f} \mathbb{E}_{\mathbf{x}\sim f}\left[\phi\big(\omega\big(r(\mathbf{x})\big)\big) + r(\mathbf{x})\tilde{\psi}\big(\omega\big(r(\mathbf{x})\big)\big)\right] \geq$$

$$\mathbb{E}_{\mathbf{x}\sim f}\left[\min_{r(\mathbf{x})\in\mathbb{L}_f}\left\{\phi\big(\omega\big(r(\mathbf{x})\big)\big) + r(\mathbf{x})\tilde{\psi}\big(\omega\big(r(\mathbf{x})\big)\big)\right\}\right] \geq$$

$$\mathbb{E}_{\mathbf{x}\sim f}\left[\min_{r}\left\{\phi\big(\omega(r)\big) + r\tilde{\psi}\big(\omega(r)\big)\right\}\right] =$$

$$\phi\big(\omega(1)\big)$$

with the last inequality being true since the minimization that follows is unconstrained and the last equality being a consequence of Theorem 1. The final lower bound is clearly attained by $r(X) = 1$, which is also a legitimate solution of the constrained minimization, since $r(X) = 1$ belongs to the class $\mathbb{L}_f$ of likelihood ratios. Consequently $r(X) = 1$ is the solution to the min-max problem. Returning to the original min-max setup with $\psi(z)$ replacing $\tilde{\psi}(z)$, we can clearly see that it satisfies equation 11. This completes the proof. ∎

## A.2 THE SUBCLASSES OF THE GANS FAMILY

**Subclass A**: $\omega(r) = r^\alpha$ The first examined subclass is the simplest one, consisting of just powers of the likelihood ratio. To the best of our knowledge, this is the first work proposing objective functions from this class. To find the pairs $\{\phi(z), \psi(z)\}$ we proceed as follows.

We have that $\omega^{-1}(z) = z^{\frac{1}{\alpha}}$ and $\mathbb{J}_\omega = [0, \infty)$. According to equation 8, for $z \in [0, \infty)$ we must define $\phi'(z) = -z^{\frac{1}{\alpha}}\rho(z), \quad \psi'(z) = \rho(z)$. The following examples can be shown to satisfy these equations.

A1) If we select $\rho(z) = z^\beta$, with $\beta \neq -1, -1 - \frac{1}{\alpha}$, this yields $\phi(z) = -\frac{z^{1+\frac{1}{\alpha}+\beta}}{1+\frac{1}{\alpha}+\beta}$ and $\psi(z) = \frac{z^{1+\beta}}{1+\beta}$. For $\beta = -1$, $\rho(z) = z^{-1}$, $\phi(z) = -\alpha z^{\frac{1}{\alpha}}$, $\psi(z) = \log z$. For $\beta = -1 - \frac{1}{\alpha}$, $\rho(z) = z^{-1-\frac{1}{\alpha}}$, $\phi(z) = -\log z$, $\psi(z) = -\alpha z^{-\frac{1}{\alpha}}$.

A2) If we select $\alpha = 1$, $\rho(z) = \frac{1}{(1+z)}$ then, $\phi(z) = -(1+z)$ and $\psi(z) = -(1+z^{-1})$.

A3) If we select $\alpha = 1$, $\rho(z) = \frac{1}{(1+z)z}$ then, $\phi(z) = -\log(1+z)$ and $\psi(z) = -\log(1+z^{-1})$.

For the particular selection $\omega(r) = r$ (corresponding to $\alpha = 1$) we can show that the resulting cost is equivalent to the Bregman cost (Bregman, 1967). In fact there is a one-to-one correspondence between our $\rho(z)$ function and the function that defines the Bregman cost. This correspondence however is lost once we switch to a different $\alpha$ or a different $\omega(r)$ function, suggesting that the proposed class of pairs $\{\phi(z), \psi(z)\}$, is far richer than the class induced by the Bregman cost.

We should mention that in A1) the selection $\alpha = 1, \beta = 0$ is known as the mean square error criterion and if we apply only the maximization problem then this corresponds to a likelihood ratio estimation technique proposed in the literature by Sugiyama et al. (2010; 2012). We will refer to this case as the *MSE* GAN.

**Subclass B**: $\omega(r) = \alpha^{-1}\log r$ This subclass considers one of the most popular transformations of the likelihood ratio, the log-likelihood ratio. As in the first subclass, for the first time, the next examples are presented. They can be used either under a min-max setting, for the determination of the generator/discriminator pair, or under a pure maximization setting for the direct estimation of the log-likelihood ratio function $\log r(\mathbf{x})$.

We have $\omega^{-1}(z) = e^{\alpha z}$ and $\mathbb{J}_\omega = \mathbb{R}$. As before $\rho(z)$ must be strictly positive and, according to equation 8, for all real $z$ we must define $\phi'(z) = -e^{\alpha z}\rho(z), \quad \psi'(z) = \rho(z)$. The following examples satisfy these equations.

B1) If $\rho(z) = e^{-\beta z}$ with $\beta \neq 0, \alpha$, this produces $\phi(z) = -\frac{e^{(\alpha - \beta)z}}{\alpha - \beta}$, $\psi(z) = -\frac{e^{-\beta z}}{\beta}$. If $\beta = 0$ then $\rho(z) = 1$, $\phi(z) = -\frac{e^{\alpha z}}{\alpha}$, $\psi(z) = z$. If $\beta = \alpha$ then $\rho(z) = e^{-\alpha z}$, $\phi(z) = -z$ and $\psi(z) = -\frac{e^{-\alpha z}}{\alpha}$. We call the $\alpha = 1, \beta = 0.5$ case the *Exponential* GAN.

B2) If $\alpha = 1$, $\rho(z) = \frac{1}{1 + e^z}$ then, $\phi(z) = -\log(1 + e^z)$ and $\psi(z) = -\log(1 + e^{-z})$.

**Subclass C**: $\omega(r) = \frac{r}{r+1}$ As we already mentioned, this is another important transform of the likelihood ratio. Interestingly, in this subclass belongs the first introduced GAN (Goodfellow et al., 2014) the *Cross Entropy* GAN.

When $\omega(r) = \frac{r}{r+1}$ we have $\omega^{-1}(z) = \frac{z}{1-z}$ and $\mathbb{J}_\omega = [0, 1]$. For $\rho(z) > 0, z \in [0, 1]$ we must define the functions $\phi(z), \psi(z)$ according to equation 8 $\phi'(z) = -\frac{z}{1-z}\rho(z)$, $\psi'(z) = \rho(z)$. The next set of examples can be seen to satisfy these equations.

C1) If we select $\rho(z) = \frac{1}{z}$, this yields $\phi(z) = \log(1 - z)$ and $\psi(z) = \log z$.

C2) Selecting $\rho(z) = (1-z)^\alpha$, with $\alpha \neq 0, -1$, yields $\phi(z) = -\frac{1}{1+\alpha}(1-z)^{\alpha+1} + \frac{1}{\alpha}(1-z)^\alpha$ and $\psi(z) = -\frac{1}{1+\alpha}(1-z)^{1+\alpha}$. For $\alpha = 0$, we have $\rho(z) = 1$ and $\phi(z) = z + \log(1-z)$, $\psi(z) = z$, while for $\alpha = -1$ we have $\rho(z) = \frac{1}{1-z}$ and $\phi(z) = -\log(1-z) - \frac{1}{1-z}$, $\psi(z) = -\log(1-z)$.

In C1) we recognize the functions used in the original article by Goodfellow et al. (2014). C2) appears for the first time.

**Subclass D**: $\omega(r) = \text{sign}(\log r)$ This is a special case of $\omega(r)$ with the corresponding function not being strictly increasing. It turns out that we can still come up with optimization problems, two of which are known and used in practice, by considering $\omega(r)$ as a *limit* of a sequence of strictly increasing functions.

*Monotone Loss:* As a first approximation we propose $\text{sign}(z) \approx \tanh(\frac{c}{2}z)$ where $c > 0$ a parameter. We note that $\lim_{c \to \infty} \tanh(\frac{c}{2}z) = \text{sign}(z)$. Using this approximation we can write

$$\text{sign}(\log r) \approx \tanh\left(\frac{c}{2}\log r\right) = \frac{r^c - 1}{r^c + 1} = \omega(r). \tag{17}$$

As we mentioned, we have exact equality for $c \to \infty$. Let us perform our analysis by assuming that $c$ is finite. We note that $\omega^{-1}(z) = \left(\frac{1+z}{1-z}\right)^{\frac{1}{c}}$ and $\mathbb{J}_\omega = [-1, 1]$. Consequently, if $\rho(z) > 0$ for $z \in [-1, 1]$, we must define $\phi'(z) = -\left(\frac{1+z}{1-z}\right)^{\frac{1}{c}}\rho(z)$, $\psi'(z) = \rho(z)$.

D1) If we let $c \to \infty$ in order to converge to the desired sign function, this yields $\phi'(z) = -\rho(z)$ and $\psi'(z) = \rho(z)$. This suggests that $\phi(z) = -\int^z \rho(x)dx$ is decreasing and $\psi(z) = \int^z \rho(x)dx = -\phi(z)$ is increasing. In fact any strictly increasing function $\psi(z)$ can be adopted provided we select $\phi(z) = -\psi(z)$.

There is a popular combination that falls under Case D1). In particular, the selection $\psi(z) = z = -\phi(z)$ reminds us of Wasserstein GAN Martin Arjovsky & Bottou (2017), with two differences, in our case $z$ should lie in $[-1, 1]$ and the discriminator is not constrained to be a Lipschitz function.

*Hinge Loss:* As a second approximation we use the expression $\text{sign}(z) \approx \text{sign}(z)|z|^{\frac{1}{c}}$, $c > 0$, which is strictly increasing, continuous and converges to $\text{sgn}(z)$ as $c \to \infty$. This suggests that

$$\text{sign}(\log r) \approx \text{sign}(\log r)|\log r|^{\frac{1}{c}} = \omega(r), \tag{18}$$

and $\omega^{-1}(z) = e^{z^c}$. Since $\omega(r)$ can assume any real value we conclude that $\mathbb{J}_\omega = \mathbb{R}$ which, clearly, differs from the previous approximation where we had $\mathbb{J}_\omega = [-1, 1]$. If $\rho(z) > 0, z \in \mathbb{R}$ then, according to equation 8 we must define $\phi'(z) = -e^{z^c}\rho(z)$, $\psi'(z) = \rho(z)$. We present the following case that leads to a very well known pair from a completely different application.

D2) If we select $\psi'(z) = \rho(z) = \{e^{-|z|^{\frac{1}{c}}} + \mathbf{1}_{z<-1}\} > 0$ then $\phi'(z) = -e^{z^{\frac{1}{c}}}\{e^{-|z|^{\frac{1}{c}}} + \mathbf{1}_{z<-1}\}$. If we now let $c \to \infty$, we obtain the limiting form for the derivatives which become $\psi'(z) = -\mathbf{1}_{z<1}$ and $\phi'(z) = \mathbf{1}_{z>-1}$. By integrating we arrive at $\phi(z) = -\max\{1+z, 0\}$ and $\psi(z) = -\max\{1-z, 0\}$. The cost based on this particular pair is called the *hinge loss* Tang (2013) and it is very popular in binary classification where one is interested only in the maximization problem. The corresponding

method is known to exhibit an overall performance which in practice is considered among the best Rosasco et al. (2004); Janocha & Czarnecki (2017). Here, as in Zhao et al. (2016), we propose the hinge loss as a means to perform adversarial optimization for the design of the generator $G(\mathbf{x})$.

This completes our presentation of examples. However, we must emphasize, that these are only a few illustrations of possible pairs $\{\phi(z), \psi(z)\}$ one can construct. Indeed combining, as dictated by equation 8, any strictly increasing function $\omega(r)$ with any positive function $\rho(z)$ generates a legitimate pair $\{\phi(z), \psi(z)\}$ and a corresponding min-max problem equation 7 that enjoys the desired solution $r(\mathbf{x}) = 1$.

## A.3 EXPERIMENTS

In our experiments we employed the two neural networks the generator and the discriminator. For the generator, we used a four-layer neural network where the first layer is linear and the remaining deconvolutional; with ReLU activation functions between the layers except the final layer where we used a sigmoid function since the output is an image with pixel values in the range $[0, 1]$. The generator input is a standard i.i.d. normal vector with dimension 64 for MNIST and 128 for Stanford Cars, CelebA and CIFAR-10. The output of the generator is a $784 \times 1$ vector for the MNIST dataset, whereas for the other datasets it is a $3072 \times 1$ vector.

For the discriminator, we used a four-layer neural network with three convolutional layers followed by a linear layer. We applied Leaky ReLUs between the layers except for the final layer where we adopted proper functions based on the range $\mathbb{J}_\omega$. For the training of the two neural networks we applied the Adam algorithm Kingma & Ba (2014) with $\beta_1 = 0.5, \beta_2 = 0.9$, learning rate $10^{-4}$ and batch size 50 for MNIST and 128 for Stanford Cars, CelebA and CIFAR-10. For all datasets, the training lasted 180000 iterations.

## A.4 CELEBA

In Figures 2, 3, 4, 5 we can see the estimation of likelihood ratio from the discriminator neural network and the FID, KID scores of the generated images during training. For every optimization problem, we tested its performance for different values of the regularization parameter $\lambda$ of the maximum gradient penalty. The different values of $\lambda$ where $\{0.01, 0.1, 1, 10\}$ as in Zhou et al. (2019). We must mention that in the cases where some $\lambda$ value is missing means that the algorithm diverged during training.

From our simulations, we notice that, especially in the case of the KID score, the likelihood ratio approximation accuracy of the discriminator coincides with the quality of the synthetic images produced by the generator. For instance, in Figure 2 for A3, $\lambda = 10$ at $\sim 150000$ iterations the likelihood ratio (which should converge to 1) starts to be more variant around the optimal value, then we notice the KID, FID scores increase indicating that the generated images quality drops. It is apparent in Figure 11 where we see some blurry, almost faceless, images. Also, for A1b, B1a, B1b, Exponential, B2 GANs, we notice the same behavior.

Moreover, in all cases, we notice a faster convergence for larger values of $\lambda$. Interestingly, there is a similar behavior, in terms of convergence, between the different $\lambda$'s. For example, in B2, for $\lambda = 10$, we notice that the variance around one is increasing. Similarly for $\lambda = 1, \lambda = 0.1$ but more slowly in terms of iterations. Therefore, it seems that the regularization parameter cannot fix the divergence problems of a GAN, but they can influence the period they will become evident.

As we already discussed, in the case of Hinge and Wasserstein GANs, the likelihood ratio value is not computable. For this reason in Figure 5 we present the discriminator output for dataset $(D(X))$ and for synthetic $(D(G(Z)))$ samples. In the Hinge GAN, the discriminator output gradually increases its range of values for $\lambda = 10$ around 130000 iteration and for $\lambda = 1$ around 200000 iteration. Accordingly, the synthetic images produced by the generator start to look less similar to the ones from the dataset. Similarly, Wasserstein GANs have the same patterns. But in this case, the discriminator output tends to have large values. Our simulations argue, that a way to reduce those large values is to increase the regularization parameter $\lambda$, in other words to "force" the discriminator being Lipschitz.

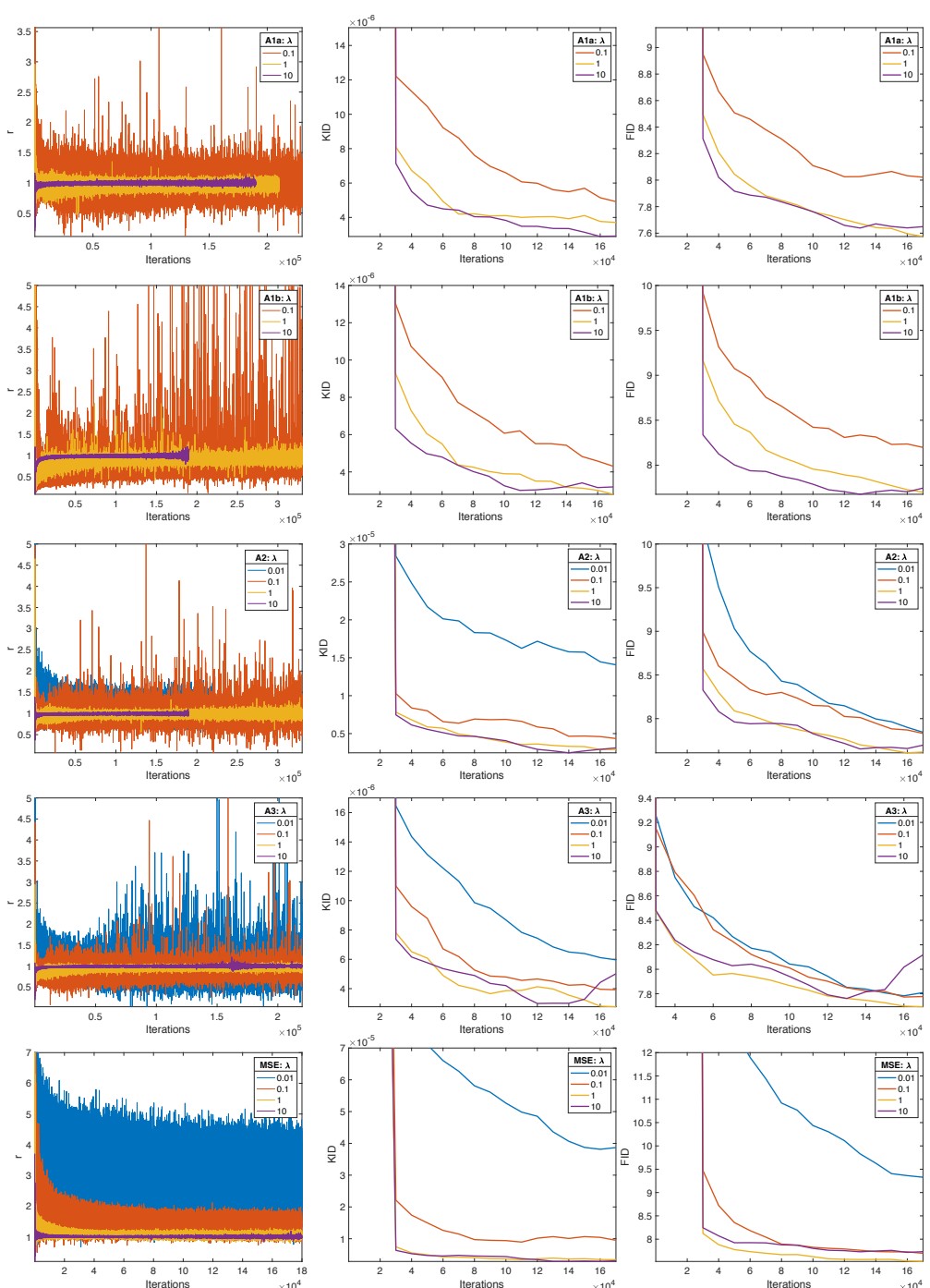

Figure 2: Likelihood ratio ($r$), KID, FID scores for CelebA dataset. Simulation results correspond to Subclass A GANs for different values of the maximum gradient penalty hyperparameter $\lambda$.

## A.5 STANFORD CARS

In Figures 6, 7, 8, 9 we can see the estimation of likelihood ratio from the discriminator neural network and the FID, KID scores of the generated images during training. For every optimization problem, we tested its performance for different values of the regularization parameter $\lambda$ of the maximum gradient penalty. The different values of $\lambda$ where $\{0.01, 0.1, 1, 10\}$ as in Zhou et al. (2019). Cases where some $\lambda$ value is missing means that the algorithm diverged during training.

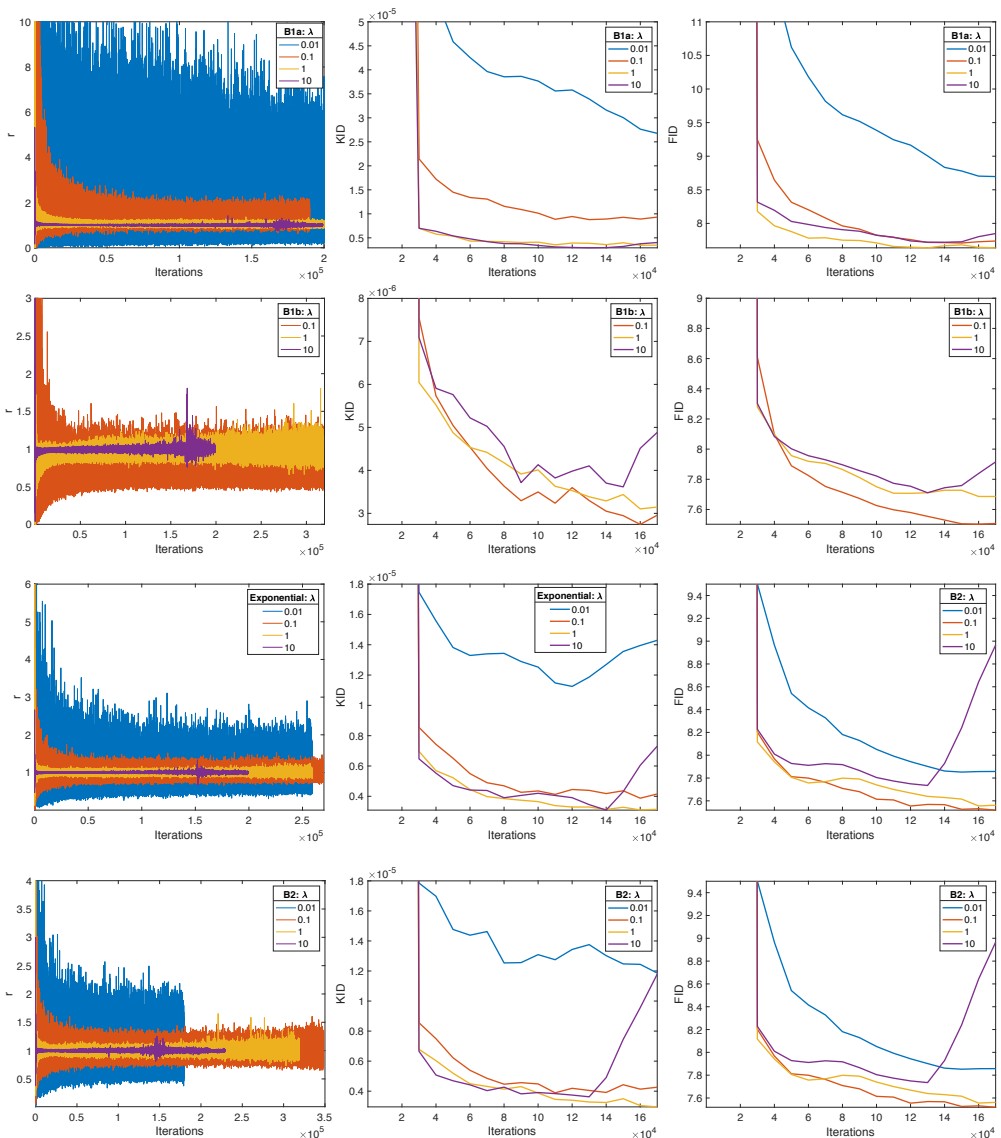

Figure 3: Likelihood ratio ($r$), KID, FID scores for CelebA dataset. Simulation results correspond to Subclass B GANs for different values of the maximum gradient penalty hyperparameter $\lambda$.

## A.6 SYNTHETIC EXAMPLES

In the Figures 10, 11 and 12 we present some random samples of the trained generators for the datasets Stanford cars, CelebA, and CIFAR-10 for $\lambda = 10$.

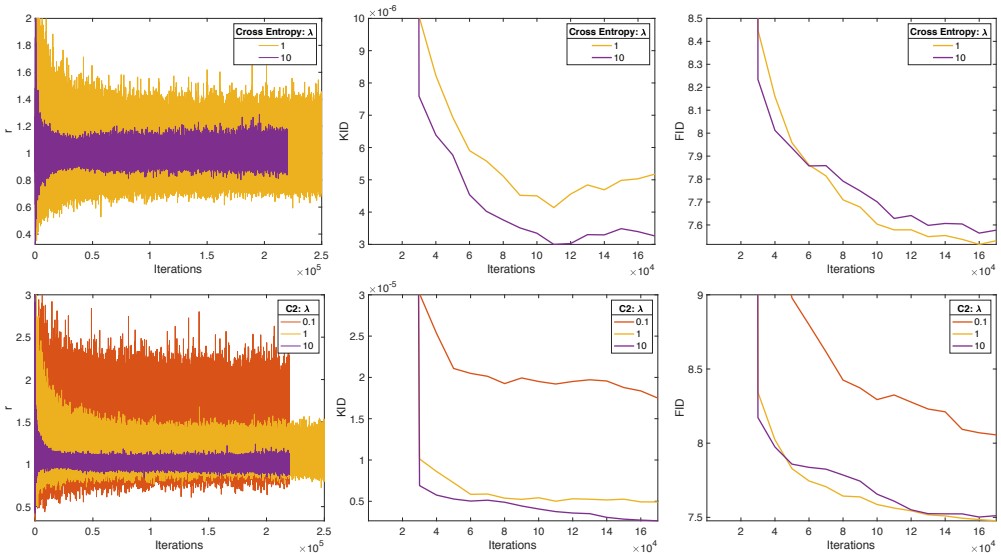

Figure 4: Likelihood ratio ($r$), KID, FID scores for CelebA dataset. Simulation results correspond to Subclass C GANs for different values of the maximum gradient penalty hyperparameter $\lambda$.

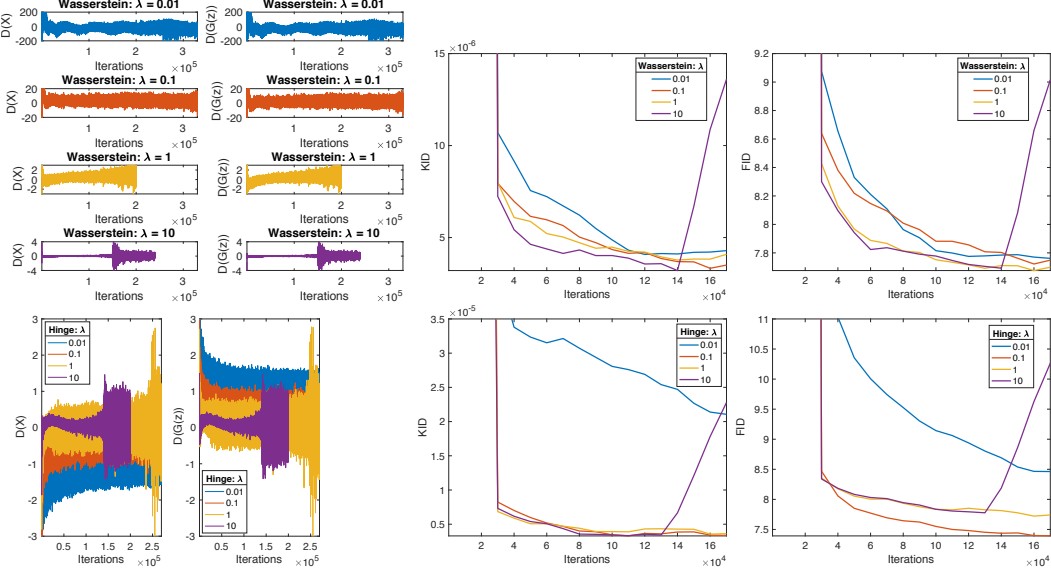

Figure 5: Discriminator outputs for real ($D(X)$) and sythetic ($D(G(Z))$) samples, KID, FID scores for CelebA dataset. Simulation results correspond to Hinge and Wasserstein GANs for different values of the maximum gradient penalty hyperparameter $\lambda$.

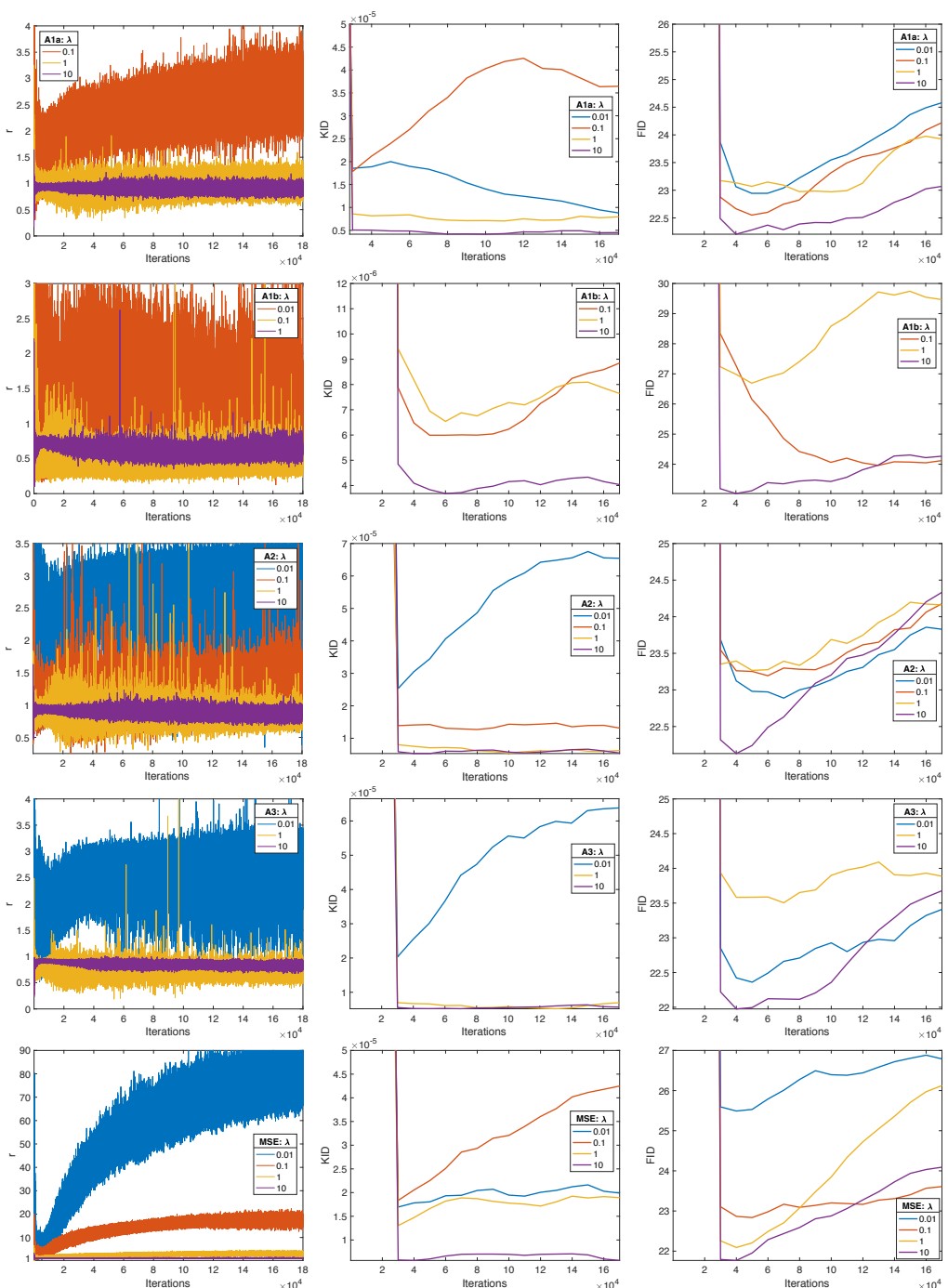

Figure 6: Likelihood ratio ($r$), KID, FID scores for Stanford cars dataset. Simulation results correspond to Subclass A GANs for different values of the maximum gradient penalty hyperparameter $\lambda$.

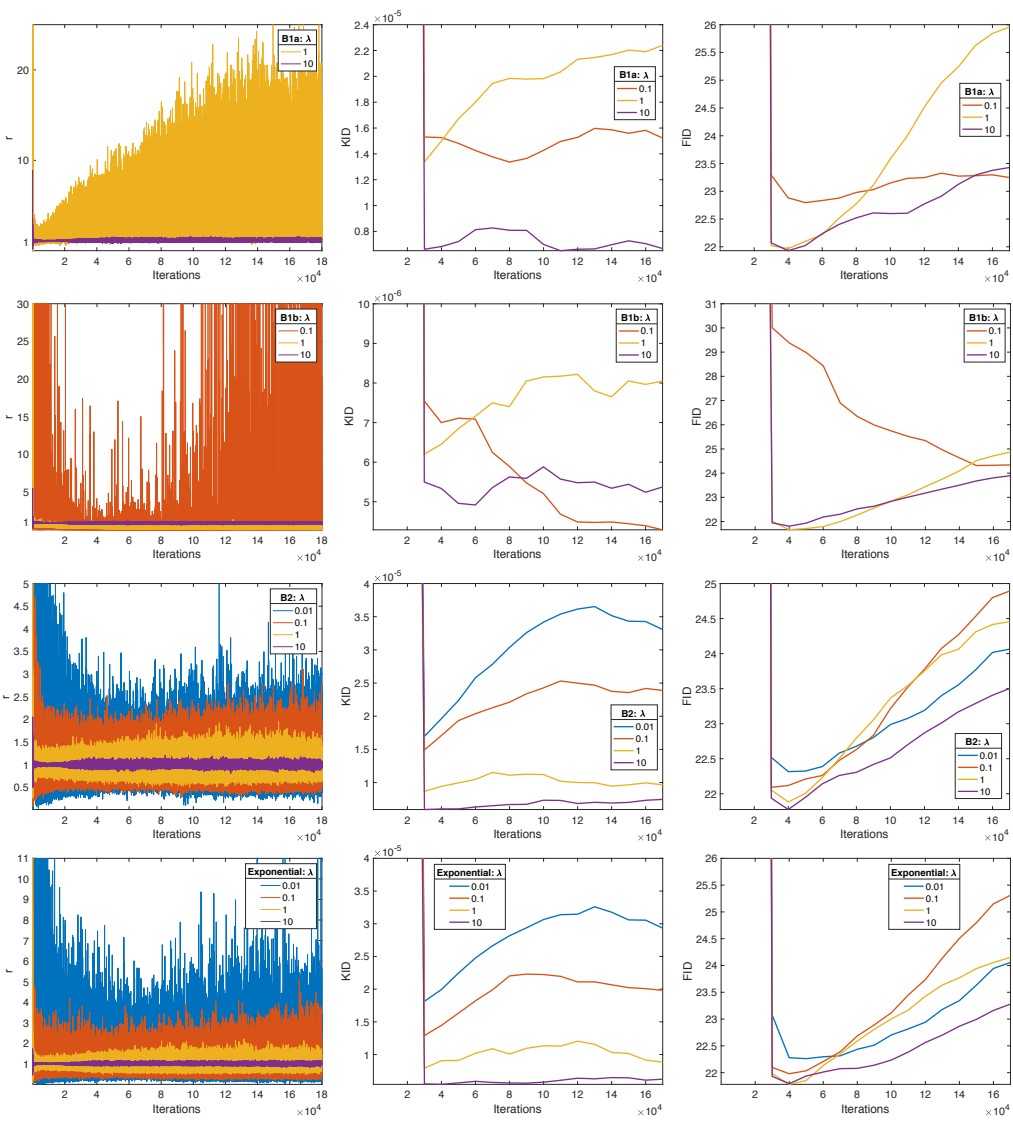

Figure 7: Likelihood ratio ($r$), KID, FID scores for Stanford cars dataset. Simulation results correspond to Subclass B GANs for different values of the maximum gradient penalty hyperparameter $\lambda$.

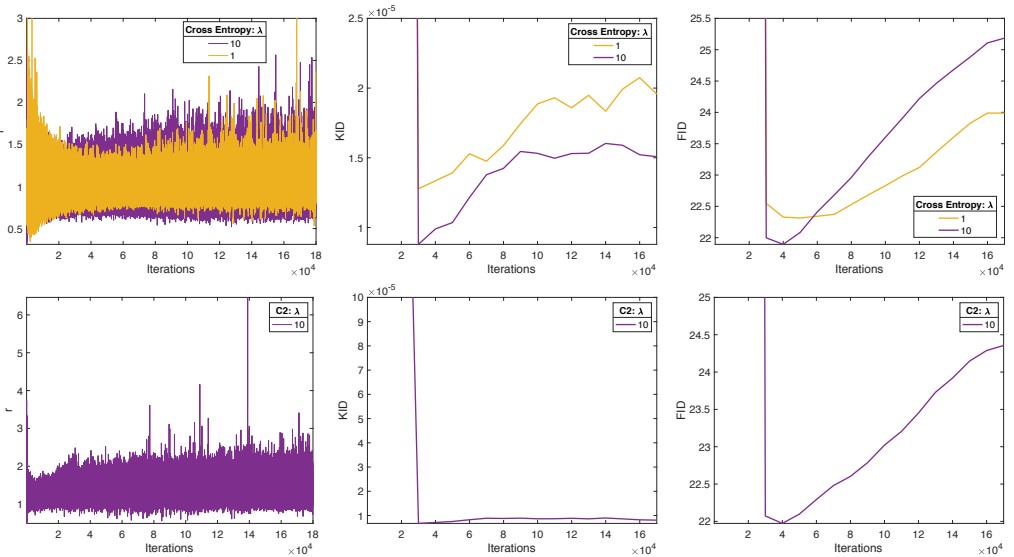

Figure 8: Likelihood ratio ($r$), KID, FID scores for Stanford cars dataset. Simulation results correspond to Subclass C GANs for different values of the maximum gradient penalty hyperparameter $\lambda$.

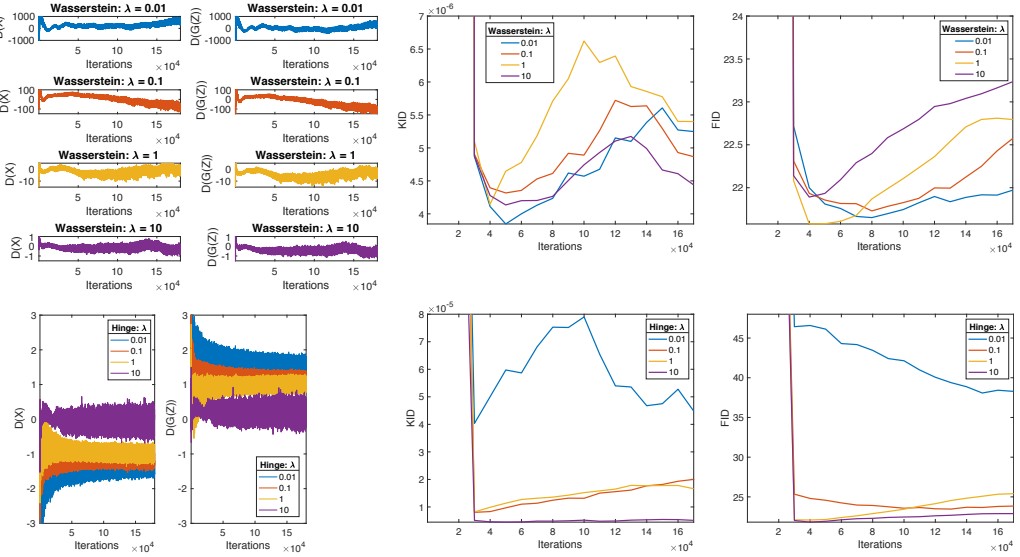

Figure 9: Discriminator outputs for real ($D(X)$) and sythetic ($D(G(Z))$) samples, KID, FID scores for Stanford cars dataset. Simulation results correspond to Hinge and Wasserstein GANs for different values of the maximum gradient penalty hyperparameter $\lambda$.

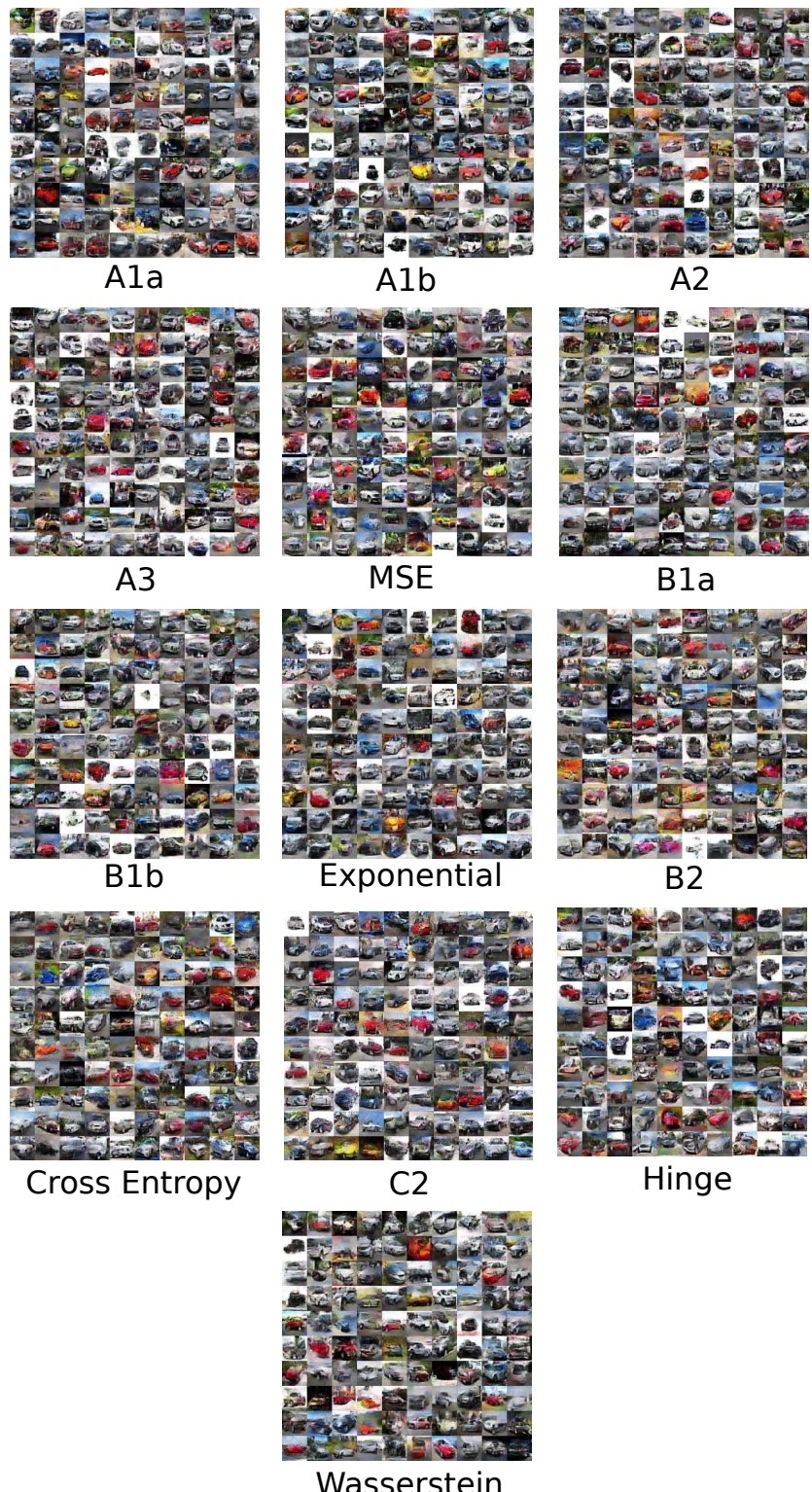

Figure 10: Random samples for different GANs objective function (and maximum gradient penalty regularization parameter $\lambda = 10$) on Stanford cars dataset.

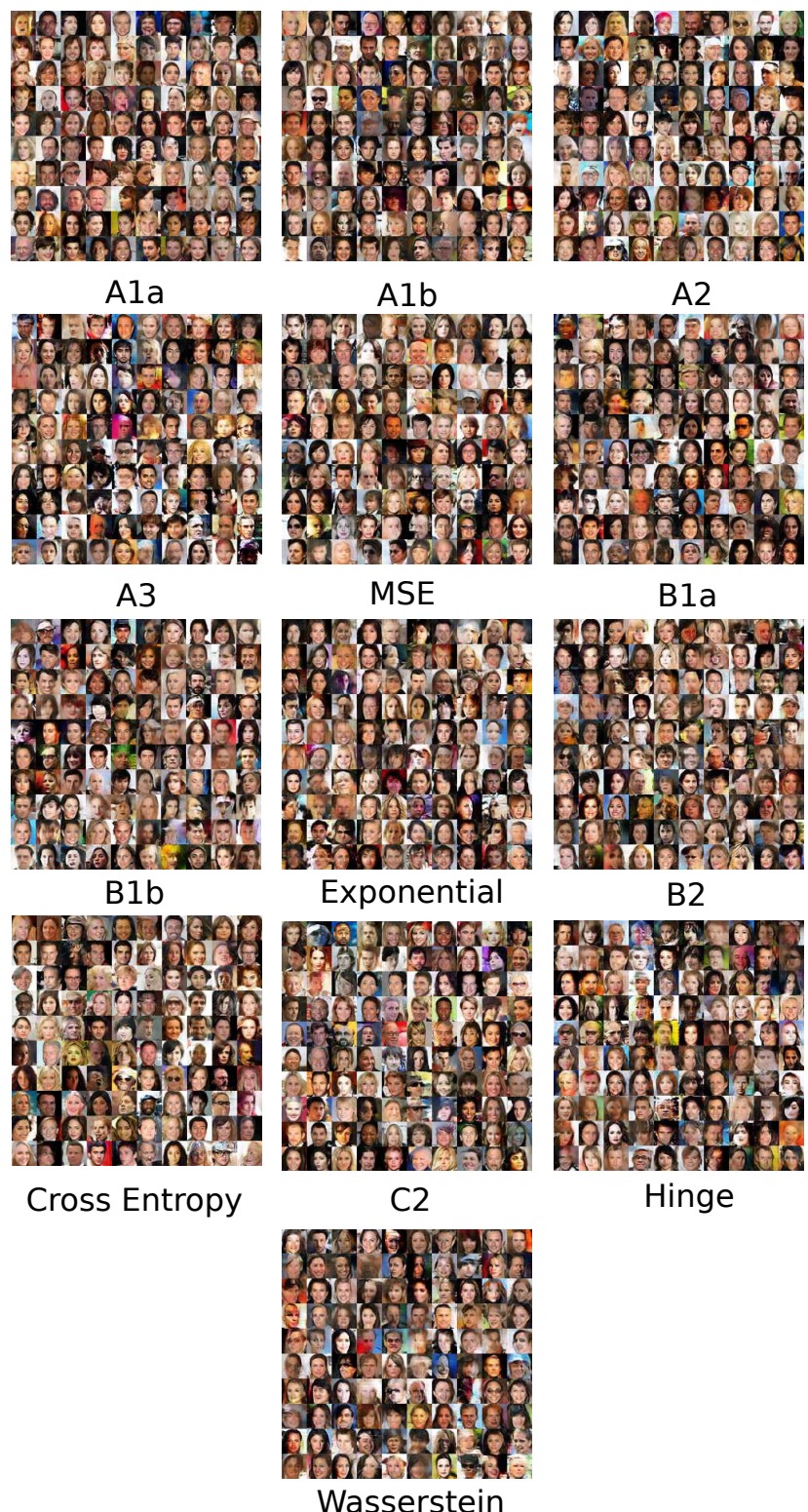

Figure 11: Random samples for different GANs objective function (and maximum gradient penalty regularization parameter $\lambda = 10$) on CelebA dataset.

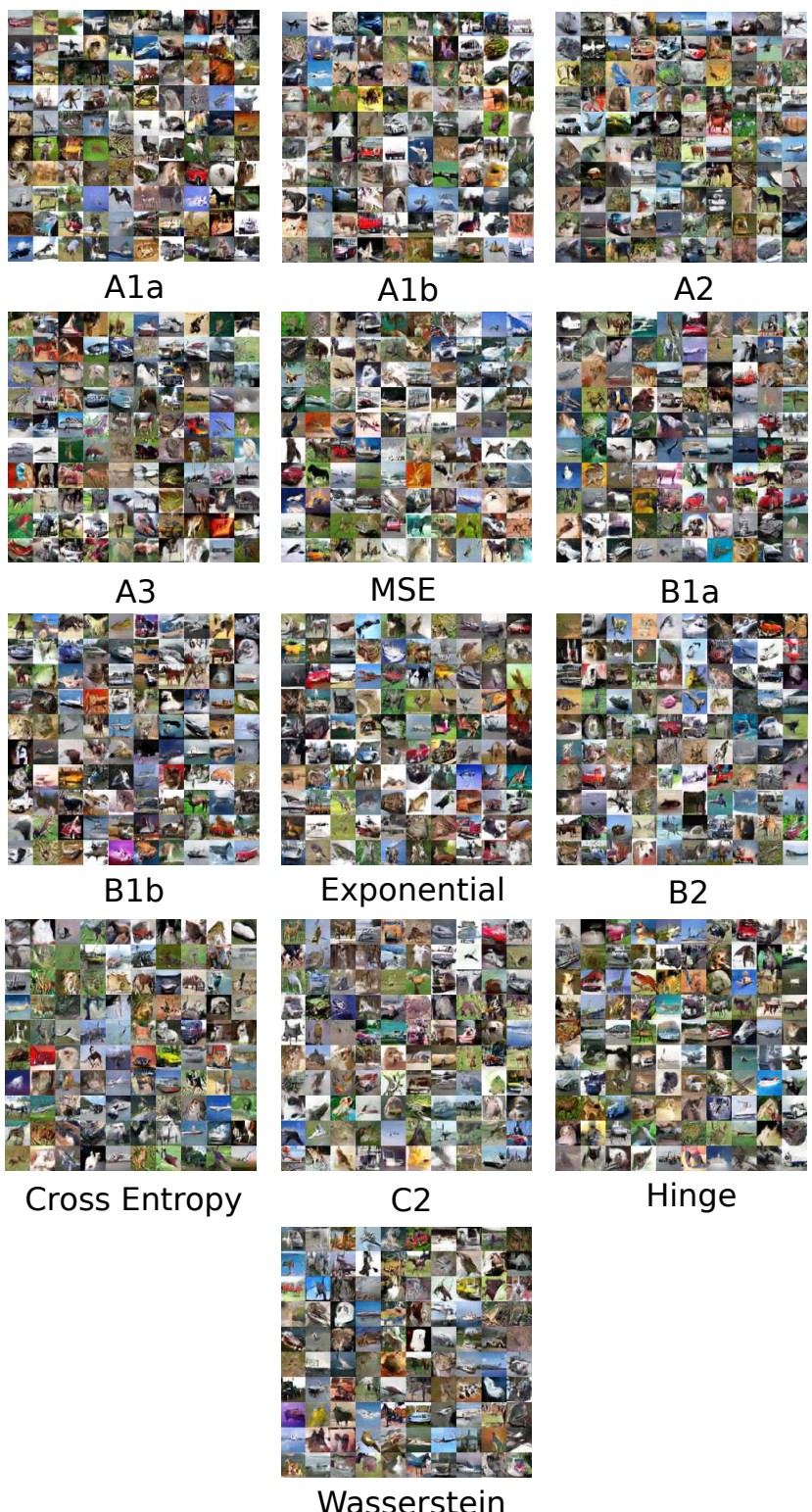

Figure 12: Random samples for different GANs objective function (and maximum gradient penalty regularization parameter $\lambda = 10$) on CIFAR-10 dataset.

