# OpenReview forum: "Adversarial Problems for Generative Networks"
_ICLR.cc/2021/Conference — Reject_

### Official Review · AnonReviewer1 · 2020-10-23
**Impactful paper for further improvements on generative models with adversarial optimization problems.**

**Rating:** 7
**Confidence:** 3

**Review:**

Overall, this paper provides impacts on understanding the core of generative models with adversarial optimization problems.
This paper shows the diverse possibilities of formulating the generative model optimization problems that the researchers can further investigate for better performances.
Also, this paper shows that generative models with unexplored losses achieve the best results in various datasets which demonstrates the possibilities of future improvements of generative models.
Overall, this paper is valuable to the machine learning community (especially for generative models and adversarial training).
The below are some concerns for this paper but those concerns are not bigger than the advantages of this paper.

1. Quantitative experiments
- Although the authors provided two tables (Table 2 and 3), there were not much analyses about the results.
- I understand that it is not an easy problem to understand "when" should we use "which" function. However, it would be great if the authors can discover some trends in the results to demonstrate which type of functions work well with which type of datasets.
- I think it would be great to use some synthetic data with known characteristics of distributions as the target distribution to analyze for understanding this point.

2. Other types of dataset
- Generative models are widely utilized in computer vision.
- However, there are various other types of datasets that can get benefits of generative models such as tabular data and time-series data.
- It would be good if the authors can provide some simple experiments to demonstrate its generalizability.

3. Minor points
- It is not clear to transform between equation (3) and (4). I think this is a critical part in this paper; thus, it would be good to explain a little bit more for this part.
- The authors explain the differences between f-GAN and this paper. However, it is not super clear to understand. It would be good to clarify this point to highlight the novelty of this paper.

--------------------------After reading other reviews are rebuttals---------------------

After reading all the reviews from other reviewers and corresponding rebuttals, I think this paper is a good paper and enough to be accepted in ICLR.
1. I think it has a clear difference from f-GAN. It can provide a new loss function for the generative models which can further extend the success of generative models in the future.
2. Experiments are not super interesting but at least it has some intuitions corresponding to the authors' claims.
3. General theoretical results for the generative models (such as when should we use which loss) is a very difficult problem to solve. Maybe this paper can provide some intuitions for solving that large problem. But it seems too much to ask this thing to the authors of this paper. Without that, I think this paper is still worth to present to the ICLR readers and participants.

Therefore, I am standing on my original score (7).

---

> ### Author Response · Authors · 2020-11-25
> **Thank you for your review**
>
> We performed some additional experiments included in the Appendix. In them, we found a relation of the quality of the synthetic data with the convergence of the estimated likelihood ratio to the optimal value. In the cases where the estimated mean of the likelihood ratio is very close to one (the optimal), a sudden increase of the variance around its mean is related to images not close to the dataset ones.
> Due to time limits, we were not able to include additional experiments for other types of images.
> Finally, we included the steps for the derivation of eq. (4) from (3).

---

### Official Review · AnonReviewer4 · 2020-10-27
**Interesting ideas, relationship to previous work unclear**

**Rating:** 4
**Confidence:** 3

**Review:**

Summary
========
In this paper, the authors set out to find what scalar functions will make for a “max” part of the “min-max” GAN objective. They then find such a class of functions, and show that only a ratio between two equal probabilities will be admitted as a solution.

Pros:
====
The paper nicely introduces a different way of seeing GANs, not as a difference between the generated and real data, but as a an integer of the ratio between generated and real distribution times the discriminator. Only if the ratio is 1 everywhere is the discriminator unable to maximize the max part of the GAN objective.

Further, I liked the idea that the discriminator shouldn’t just decide what class data belongs to, but also estimate the probability ratio. Specifically, in the formulation here, the max part is maximized when $D(X) =\omega(r(X))$, so maximized iff $\omega^{-1}(D(x))$ doesn’t just classify, but says the probability ratio between the two classes. If this idea is expanded upon, I think the authors could make a novel contribution.

Cons:
=====

Unfortunately, the authors have neglected to carefully explain how their contribution relates to previous work. It’s telling that the paper cites only two papers from 2018, one from 2019 and none from 2020. All other citations are from previous years, even though 2018-2020 has been a time of much GAN research.

A key way in which the author’s work hasn’t been sufficiently compared to previous work is with their main claim “We propose a simple methodology for constructing such [min-max] problems assuring, at the same time, consistency of the corresponding solution.” In [Liu], they show a class of of functions where consistency is also guaranteed, and the class shown by the authors here is a subset of the class in [Liu]. The details are at the bottom of my review

Further, many of the techniques in this paper seem very similar to [Song], where they also investigate the f*-gan divergence. Specifically, the claims they make in Theorem 1 seem very similar to Prop. 2 in [Song]. Also the change of measure trick in the introduction can be found in [Song]. A detailed comparison of this work to that work would also be helpful.

Since when reading this paper one simply doesn’t know what is previous work which has already been done by others and what is the author’s novel contribution. Once the authors address this, and one is confident the contribution is indeed novel, then the submission would be worth considering.

Details of why this is a subset of what’s already been shown in [Liu]:

There, they examine the difference between the target density $d$ (in this paper $d$ is $f$, but Liu uses $f$ for something else) and the generated density $g$ via $\sup_{f\in\mathcal F}\mathbb E_{x\sim d,y\sim g}[f(x,y)]$, so we find the function $f$ in a class $\mathcal F$ which maximally separates the classes from $d$ and $g$.

Now this work proposes to do the same thing, but with $f(x,y)=\phi(D(x)) - \psi(D(y))$ where $\phi(z) = -\int_{\omega^{-1}(0)}^z \omega^{-1}(t)p(t) dt + C_1 $ and $\psi(z)=\int_{\omega^{-1}(0)}^z p(t) dt + C_2$.

In [Liu] they then split f(x,y) up into two functions m and r, such that f(x,y)=m(x, y) - r(x,y) where m(x,y) has the form m(x,y)=v(x)-v(y). This can be done in your case too, resulting in (here we drop the constants C_1 and C_2 for simplicity)

$v(x) = \int_{\omega^{-1}(0)}^{D(x)} p(t) dt$, $v(y) = \int_{\omega^{-1}(0)}^{D(y)} p(t) dt$

and

$r(x,y) = \int_{\omega^{-1}(0)}^{D(x)} (\omega^{-1}(t) + 1) p(t)dt$

Since D(x) must be in $\mathcal J_\omega$, this integral has an infimum, and theorem 4 from [Liu] can be applied to achieve the same results as in this paper.

[Song] Song, Jiaming, and Stefano Ermon. "Bridging the Gap Between $ f $-GANs and Wasserstein GANs." arXiv preprint arXiv:1910.09779 (2019).

[Liu] Liu, Shuang, Olivier Bousquet, and Kamalika Chaudhuri. "Approximation and convergence properties of generative adversarial learning." Advances in Neural Information Processing Systems. 2017.

---

> ### Author Response · Authors · 2020-11-25
> **Thank you for your review, our prior work is updated**
>
> Regarding the prior work, we added more papers in the Introduction section. Also, we included Liu et. al. and Song. et. al. For the second paper the proposition 2 is not related to our Theorem 1. In Song et. al. this proposition is showing how WGANs are derived from their proposed generalized class $L^R_f(T; P, Q)$. But our Theorem 1 along with
> Corollary 1 shows that our class of optimization problems has a solution to the maximization problem, that the discriminator function approximates a transformation of the likelihood ratio, so the minimization part gives the desired result that the likelihood ratio between the data pdf and the generator output pdf is equal to one. But we include it in the prior work as a generalized family of $f$-gans.
>
> So, we believe that our work is important because our point of view, based on what quantity the discriminator function is trying to approximate, comes with a straightforward methodology of selecting different GAN losses. On the other hand, the divergence-based formulations include an extra optimization step for selecting each GAN loss. This benefit of our approach allows us to present a myriad of new GANs optimization problems. Moreover, in our case, we can choose which transformation of the likelihood ratio the discriminator function has to approximate. This is important because the neural network model (approximating the discriminator function) might have practical difficulty to estimate a transformation $\omega_1(r)$ over $\omega_2(r)$.
>
> Finally, when $\omega(r)$ is invertible, we can compute the likelihood ratio as a function of the discriminator function. This will allow us to compare each GAN loss (with such $\omega(r)$) to how close they converge to the optimal value of the likelihood ratio.

---

### Official Review · AnonReviewer3 · 2020-10-28
**The theoretical results are good, but not surprising**

**Rating:** 6
**Confidence:** 4

**Review:**

This paper generalizes the min-max problem of GANs to form a richer family of generative adversarial networks. Interestingly, most of the well-known variants of GANs can be found in the spectrum of formulations covered by the family proposed in this work.
In terms of modeling, it is evident that the family proposed in the paper is richer than that of f-GAN. The family in this paper is shown to have a connection to WGAN except that the Lipschitz condition is omitted.
However, under the light of existing works including f-GAN and other relevant works, the obtained theoretical results are not surprising to me. In addition, apart from providing a richer family, this work does not significantly influence the practical aspects of GANs.
I have some following questions:
1. If we solve the min-max problem in (2) subjected the fact that \phi and \psi satisfy Eq. (9), is it equivalent to minimizing any divergence between two distributions with pdfs f and g?
2. D(x) is not a typical discriminator whose values between [0;1] providing the probability to distinguish true and fake data, is not it? D is more similar to a critique whose output values are real-valued, is not it?

---

> ### Author Response · Authors · 2020-11-25
> **Thank you for your review**
>
>   For your first question:
>
> As we mentioned in our work there is a one-to-one correspondence of our class and $f$-gans. For example the cross-entropy GAN in the $f$-gan family is related to the minimization of the Jensen-Shannon divergence, $D_{JS}$ through $2D_{JS} - \log(4)$. In it is related with the likelihood ratio transformation $\omega(r) =\frac{r}{r+1}$.
>
> For your second question:
>
>  In our formulation, the Discriminator approximates a transformation of the likelihood ratio between g and f. Therefore, in general, the discriminator output is not constrained in $[0,1]$. A typical case where the discriminator output is constrained in $[0,1]$ is the Cross-Entropy GAN, introduced by Goodfellow et. al.. In that case, the discriminator approximates the $\frac{r}{r+1} \in [0,1]$ for $r\geq 0$.
>
>
> When compared with previous works, we believe that our work is important because our point of view, which is based on what quantity the discriminator function is trying to approximate, comes with a straightforward methodology of selecting different GAN losses. On the other hand, the divergence-based formulations include an extra optimization step for selecting each GAN loss. This benefit of our approach allows us to present a myriad of new GANs optimization problems. Moreover, in our case, we can choose which transformation of the likelihood ratio the discriminator function has to approximate. This is important because the neural network model (approximating the discriminator function) might have practical difficulty to estimate a transformation $\omega_1(r)$ over $\omega_2(r)$. Finally, when $\omega(r)$ is invertible, we can compute the likelihood ratio as a function of the discriminator function. This will allow us to compare each GAN loss (with such $\omega(r)$) to how close they converge to the optimal value of the likelihood ratio.

---

### Official Review · AnonReviewer2 · 2020-10-29
**Interesting framework but with several limitations**

**Rating:** 4
**Confidence:** 4

**Review:**

**Summary of contributions:**
This paper proposes a new framework to design new loss for GANs. The authors show that their framework is quite general and encompass a number of existing approaches (e.g. the original GAN formulation, hinge loss, etc..), they also propose a categorization in three different classes and derive new loss function. They then compare experimentally the different existing loss and the new proposed loss that fall under their framework.

**Main comment**:
The framework proposed in the paper is interesting since it's quite general and the authors are able to derive a large number of existing as well as new loss from it. However, I think the framework has several limitations:
1. The formulation is based on the likelihood ratio which is only defined if the support of $g$ and $f$ match, this is known to not be the case in the context of GANs.
2. The benefit of the framework is not clear, while it provides a way to derive new loss it's not clear what are the advantages of the new loss. Theoretically the author argue that it is a hard question to answer, and I agree. The authors try to answer this question through experiments but I find the experiments not very convincing. In particular, the authors argue that subclass A objectives are more stable based on the CelebA experiment, however it's not clear to me that the instability is due to a specific choice of objective function, it might just be that the hyper parameters where slightly off for the other objectives. I believe it would be interesting to understand better the results on CelebA, in particular maybe to show that some objectives are indeed more stable, they can vary several hyper-parameters and compare how often each objective is better than the other, that would make the results and conclusion much more convincing.

*Minor comment*:
The paper is overall clear but the clarity of some sections could be improved. I think theorem 1 would be more clear if stated a bit differently simply saying that $D=\omega(r)$ maximize $\phi(D)+r \psi(D)$ and that $r=1$ minimize $\phi(\omega(r))+r \psi(\omega(r))$. Section 3 is a bit dense, the subclasses also seem a bit arbitrary. I believe section 5 could be improved by stating more clearly the different observations, right now it looks more like a description of the figures than a clear statement of the question that the experiments try to answer and how they answer it.

---

> ### Author Response · Authors · 2020-11-25
> **Thank you for your review, additional experiment results were included**
>
> For the first-mentioned limitation of our work:
>
> In generative networks the goal is the output of the generator, g, to follow the same distribution with the dataset f, therefore f=g.
> In our formulation, there are no restrictions about the supports of the two pdfs. In our simulations, we constrained the generator output to the same range as the dataset because this is the most common strategy when training GANs for images. But of course, the generator output could have a larger range from the dataset, for example, $G(z) \in \mathbb{R}$ when the dataset range is in $[0,1]$.
>
> For the second-mentioned limitation of our work:
>
> The simulation setup followed for our simulations was the same as the Gulrajani et. al. and Zhou et. al.. We performed more experiments for the three datasets CelebA, Stanford cars by changing the maximum gradient penalty regularization parameter $\lambda$ in Zhou et. al.. As in Zhou et. al. $\lambda = \{0.01,0.1,1,10\}.$ The additional experiment results are in the Appendix. Our additional simulations in the case of CelebA, which was the most unclear case, showed that the choice of $\lambda$ just postponed the divergent behavior of some specific GANs (i.e Hinge, Wasserstein, Exponential, B2).
>
> Moreover, we moved some of the details of Section 3 in the appendix.
>
>
> [Gulrajani et. al.] Ishaan Gulrajani, Faruk Ahmed, Martin Arjovsky, Vincent Dumoulin, and Aaron C Courville. Improved training of Wasserstein gans. In Advances in neural information processing systems, pp. 5767–5777, 2017.
>
> [Zhou et. al.] Zhiming Zhou, Jiadong Liang, Yuxuan Song, Lantao Yu, Hongwei Wang, Weinan Zhang, Yong Yu, and Zhihua Zhang. Lipschitz generative adversarial nets. arXiv preprint arXiv:1902.05687, 2019.

---

### Decision · Program_Chairs · 2021-01-07
**Final Decision**

**Decision:**

Reject

**Comment:**

This paper proposed a new family of losses for GANs and showed that this family is quite general and encompasses a number of existing losses as well as some new loss functions. The paper compared experimentally the existing losses and the new proposed losses. But the benefit of this family is not clear theoretically, and this work did not also provide the very helpful insights for the practical application of GANs.